# Phage-mediated peripheral kill-the-winner facilitates the maintenance of costly antibiotic resistance

Chujin Ruan [1,4] ✉, Deepthi P. Vinod[1,2,4] & David R. Johnson [1,3] ✉

The persistence of antibiotic resistant (AR) bacteria in the absence of antibiotic pressure raises a paradox regarding the fitness costs associated with antibiotic resistance. These fitness costs should slow the growth of AR bacteria and cause them to be displaced by faster-growing antibiotic sensitive (AS) counterparts. Yet, even in the absence of antibiotic pressure, slower-growing AR bacteria can persist for prolonged periods of time. Here, we demonstrate a mechanism that can explain this apparent paradox. We hypothesize that lytic phage can modulate bacterial spatial organization to facilitate the persistence of slower-growing AR bacteria. Using surface-associated growth experiments with the bacterium *Escherichia coli* in conjunction with individual-based computational simulations, we show that phage disproportionately lyse the faster-growing AS counterpart cells located at the biomass periphery via a peripheral kill-the-winner dynamic. This enables the slower-growing AR cells to persist even when they are susceptible to the same phage. This phage-mediated selection is accompanied by enhanced bacterial diversity, further emphasizing the role of phage in shaping the assembly and evolution of bacterial systems. The mechanism is potentially relevant for any antibiotic resistance genetic determinant and has tangible implications for the management of bacterial populations via phage therapy.

The rise of antibiotic resistance is an urgent threat to global public health and is causing increasing morbidity and mortality worldwide[1,2]. A broadly applied strategy to combat this threat is the judicious use and disposal of antibiotics. This strategy anticipates that antibiotic-resistant (AR) bacteria will be displaced by antibiotic-sensitive (AS) counterparts in the absence of antibiotic pressure[3,4]. This is because antibiotic resistance often imposes fitness costs that slow the growth of AR bacteria, which provides a growth advantage to AS counterparts that lose antibiotic resistance through genetic mutation or plasmid loss[3,4]. However, despite the implementation of this strategy[5–7], AR bacteria continue to persist in the absence of antibiotic pressure in a myriad of environments[8,9].

Several mechanisms have been proposed to explain how slower-growing AR bacteria can persist in the face of faster-growing AS counterparts in the absence of antibiotic pressure[10–13]. These mechanisms include compensatory mutations that reduce the fitness costs associated with antibiotic resistance or their associated genetic elements[14,15], co-selection of linked traits[16–19], and horizontal transfer of antibiotic resistance genetic determinants[20–22]. One aspect that can affect the dynamics of antibiotic resistance determinants is bacterial spatial organization[23–31]. Experiments and theoretical considerations have illustrated how spatial processes can increase the persistence of neutral and even deleterious genetic mutations at the periphery of growing biomass[32–34]. Whether these processes can explain the

[1]Department of Environmental Microbiology, Swiss Federal Institute of Aquatic Science and Technology (Eawag), Dübendorf, Switzerland. [2]Department of Environmental Systems Science, Swiss Federal Institute of Technology (ETH), Zürich, Switzerland. [3]Institute of Ecology and Evolution, University of Bern, Bern, Switzerland. [4]These authors contributed equally: Chujin Ruan, Deepthi P. Vinod. ✉e-mail: chujin.ruan@eawag.ch; david.johnson@eawag.ch

persistence of slower-growing AR bacteria in the face of faster-growing AS counterparts, however, remains unclear. Because surface-associated bacterial systems are important reservoirs of antibiotic resistance genetic determinants[35], understanding how bacterial spatial organization and dynamics affect the persistence of antibiotic resistance could set the stage for developing more effective bacterial management and intervention strategies.

We hypothesize here that phage lysis can modulate bacterial spatial organization to increase the persistence of slower-growing AR bacteria in the face of faster-growing AS counterparts. More precisely, we hypothesize that lytic phage can mediate a peripheral kill-the-winner dynamic; faster-growing AS counterpart cells will be disproportionally lysed to a greater extent than slower-growing AR cells, consequently increasing the persistence of antibiotic resistance (Fig. 1). Our hypothesis is grounded in the fundamental principle that, for surface-associated bacterial systems, biomass growth is primarily driven by cells located at the biomass periphery where resources supplied from the environment are plentiful[36–38]. Because of their differences in growth rates, slower-growing AR bacteria will be disproportionately located behind the biomass periphery, while faster-growing AS counterpart cells will disproportionately occupy the biomass periphery (Fig. 1). Due to mass transfer limitations, phage will predominantly lyse cells located at the biomass periphery[39,40], which will disproportionately be AS counterpart cells (Fig. 1). The faster-growing AS counterpart cells will therefore undergo more vigorous phage lysis, thereby offsetting their growth advantage and increasing the persistence of slower-growing AR cells (Fig. 1).

Our hypothesis not only provides an explanation for the persistence of slower-growing AR cells in the face of faster-growing AS counterpart cells but also makes predictions regarding dynamic environments where spontaneous genetic changes that alter antibiotic resistance profiles can occur concurrently with phage lysis. These alterations in antibiotic resistance profiles can occur through genetic mutations or through plasmid loss, both of which can relieve cells of the fitness costs associated with antibiotic resistance[41–43]. After such genetic changes occur, the faster-growing AS counterpart cells that emerge will grow towards and disproportionately occupy the biomass periphery, consequently making them more susceptible to phage lysis and increasing the persistence of slower-growing AR cells. We therefore predict that, even if AR cells are capable of reverting to AS cells via spontaneous genetic changes, the peripheral kill-the-winner dynamic will rapidly establish itself to disproportionately lyse the AS cells and

promote the persistence of the slower-growing AR cells, thus providing a general mechanism for how antibiotic resistance genetic determinants can persist amidst ongoing evolutionary processes.

To test our hypothesis, we performed surface-associated growth experiments with isogenic derivatives of the bacterium *Escherichia coli* MG1655 in the absence or presence of the lytic phage T6. The parental strain, which we refer to as strain AS for antibiotic sensitive, is sensitive to all the antibiotics used in this study. We then obtained antibiotic-resistant variants of strain AS, which we refer to as AR strains for antibiotic-resistant. The AR strains contain antibiotic resistance determinants that are either single genetic mutations located on the chromosome (strain $AR_{C,Tet}$ is resistant to tetracycline while strain $AR_{C,Str}$ is resistant to streptomycin) or are genes located on the non-transmissible plasmid pEF001 (strain $AR_{P,Chl}$ is resistant to chloramphenicol) that can be spontaneously lost during cell division. We then assembled strain AS with either strain $AR_{C,Tet}$ or $AR_{C,Str}$ into co-cultures, propagated them across nutrient-amended agar surfaces in the absence of antibiotic pressure, and quantified the strain abundances and emergent spatial patterns using confocal laser scanning microscopy (CLSM). We also propagated strain $AR_{P,Chl}$ alone across nutrient-amended agar surfaces in the absence of antibiotic pressure, tracked the spontaneous emergence of plasmid-free AS cells, and quantified the emergent spatial patterns with CLSM. In both cases, we expected that the AR strains, all of which grow slower than the AS strain, would have increased persistence in the presence of phage T6 via our proposed peripheral kill-the-winner dynamic (Fig. 1). Finally, we complemented our experiments with individual-based computational simulations to identify general mechanisms for how phage lysis can increase the persistence of slower-growing AR strains.

## Results

### Phage lysis increases the persistence of slower-growing AR cells
We first quantified how phage lysis affects the persistence of slower-growing AR cells in the face of faster-growing AS counterparts. To accomplish this, we performed surface-associated growth experiments in the absence or presence of phage T6. We mixed strain AS with either strain $AR_{C,Tet}$ or $AR_{C,Str}$, both of which contain single chromosomal mutations that bestow resistance to tetracycline or streptomycin, at a 1:1 initial cell ratio. To distinguish them, strain AS expressed a green fluorescent protein-encoding gene (falsely colored to cyan in the images) located on the chromosome, while strains $AR_{C,Tet}$ and $AR_{C,Str}$

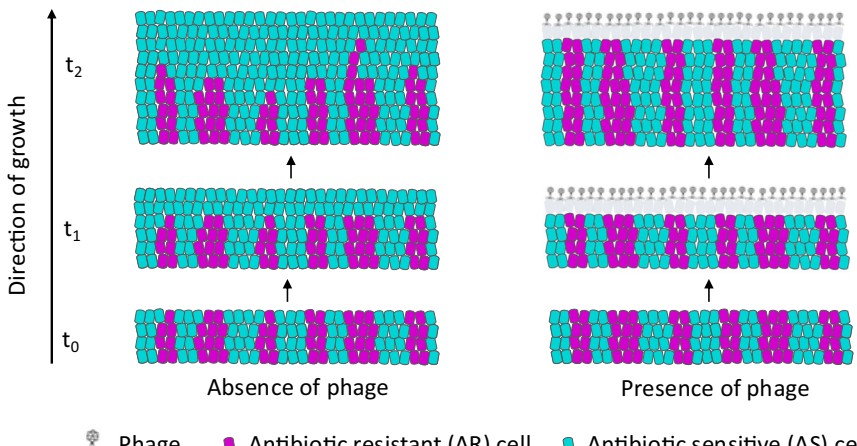

**Fig. 1 | Schematic of the peripheral kill-the-winner hypothesis.** In the absence of phage, we expect that faster-growing antibiotic-sensitive (AS) cells (cyan) will displace slower-growing antibiotic-resistant (AR) cells (magenta) along the biomass periphery. In the presence of phage, however, we expect that the slower-growing AR cells will persist with the faster-growing AS cells. This is because the faster-growing AS cells will disproportionately occupy the biomass periphery, and they will therefore be more susceptible to phage lysis. This will increase the removal of AS cells from the biomass and counteract the benefits of their faster growth relative to the slower-growing AR cells, thus increasing the persistence of the slower-growing AR cells.

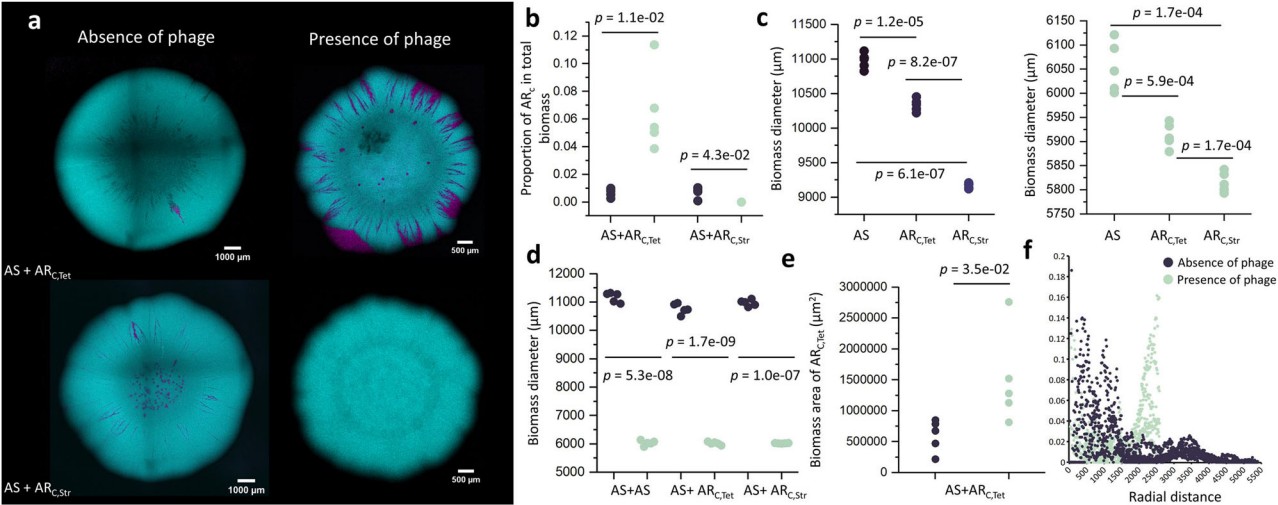

**Fig. 2 | Phage lysis increases the persistence of slower-growing AR cells.**
**a** Representative CLSM images of co-cultures of strains AS (cyan) and $AR_{C,Tet}$ (magenta) (upper images) or of strains AS (cyan) and $AR_{C,Str}$ (magenta) (lower images) in the absence or presence of phage T6. We imaged the biomass after ten days of incubation in the absence of antibiotic pressure. **b** The proportions of the total biomass areas occupied by strains $AR_{C,Tet}$ or $AR_{C,Str}$ when grown in co-culture with strain AS in the absence or presence of phage T6. **c** The biomass diameters of strains AS, $AR_{C,Tet}$ and $AR_{C,Str}$ when grown in monoculture in the absence (left) or presence (right) of phage T6. **d** The biomass diameters of co-cultures of strains AS and $AR_{C,Tet}$ or strains AS and $AR_{C,Str}$ in the absence or presence of phage T6. **e** The biomass areas (population size) of strain $AR_{C,Tet}$, when grown in co-culture with strain AS in the absence or presence of phage T6. **f** The proportions of $AR_{C,Tet}$ cells within co-cultures as a function of the radial distance from the centroid of the biomass. For **b**–**f**, each data point is an independent experimental replicate ($n = 5$), the black data points are for experiments in the absence of phage T6, and the green data points are for experiments in the presence of phage T6. For **b**–**e**, the $p$ values are for two-sample two-sided Welch tests. Source data are provided as a Source Data file.

expressed a red fluorescent protein-encoding gene (falsely colored to magenta in the images) located on the chromosome. We then grew the co-cultures across nutrient-amended agar surfaces in the absence of antibiotic pressure and quantified the patterns of spatial organization that emerged with CLSM (Fig. 2a).

In the absence of phage T6, we found that strain AS had a clear competitive advantage over both strains $AR_{C,Tet}$ and $AR_{C,Str}$ (Fig. 2a). After ten days of incubation, strains $AR_{C,Tet}$ and $AR_{C,Str}$ constituted less than 0.7% of the total biomass area even though they constituted ~50% of the initial inoculum (Fig. 2b). We attribute this effect to the fitness cost associated with the antibiotic resistance determinants for tetracycline and streptomycin. This is evident from our experiments, where the total biomass areas of strains $AR_{C,Tet}$ or $AR_{C,Str}$ when grown alone in the absence of antibiotic pressure were both significantly smaller than the area of strain AS when grown alone (two-sample two-sided Welch tests; $AR_{C,Tet}$, $p = 1.2 \times 10^{-5}$, $n = 5$; $AR_{C,Str}$, $p = 6.1 \times 10^{-7}$, $n = 5$) (Fig. 2c). We performed additional batch culture growth experiments to further corroborate these observations, confirming that both AR strains grow slower than strain AS, with strain $AR_{C,Str}$ exhibiting the highest fitness cost (two-sample two-sided Welch tests; AS vs $AR_{C,Tet}$: $p = 1.6 \times 10^{-8}$; AS vs $AR_{C,Str}$: $p = 5.3 \times 10^{-9}$; $AR_{C,Tet}$ vs $AR_{C,Str}$: $p = 6.5 \times 10^{-5}$) (Supplementary Fig. 1).

In the presence of phage T6, we found that the slower-growing strain $AR_{C,Tet}$ can persist in the face of the faster-growing strain AS (Fig. 2a). The proportion of the area occupied by strain $AR_{C,Tet}$ was nearly 10-fold larger when in the presence of phage T6 (two-sample two-sided Welch test; $p = 1.1 \times 10^{-2}$, $n = 5$) (Fig. 2b). Moreover, even though the presence of phage T6 significantly reduced the overall biomass area of all the strains (two-sample two-sided Welch tests; $p < 1.0 \times 10^{-7}$, $n = 5$) (Fig. 2d), the absolute population size of the slower-growing strain $AR_{C,Tet}$ increased (two-sample two-sided Welch test; $p = 3.5 \times 10^{-2}$, $n = 5$) (Fig. 2e). The increased persistence of the $AR_{C,Tet}$ cells could not be explained by differences in its susceptibility to phage T6. Despite their chromosomal mutations, both AR strains remained susceptible to phage T6, forming smaller biomass diameters on agar plates (Fig. 2c). Furthermore, the burst

sizes and latency periods of phage T6 also did not significantly differ among the AR strains, which provides further support that phage infection dynamics remained consistent despite the mutations (Supplementary Fig. 2). We further analyzed the spatial positionings of the $AR_{C,Tet}$ and AS cells. In the absence of phage T6, the faster-growing AS cells dominated the biomass periphery while the slower-growing $AR_{C,Tet}$ cells were positioned behind the periphery where nutrients supplied from the environment were depleted (Fig. 2f). In the presence of phage T6, in contrast, the slower-growing $AR_{C,Tet}$ cells persisted at the biomass periphery where nutrients supplied from the environment were plentiful (Fig. 2f).

In contrast with strain $AR_{C,Tet}$, we found that the slower-growing strain $AR_{C,Str}$ was unable to persist when co-cultured with the faster-growing strain AS in the absence of antibiotic pressure, regardless of whether phage T6 was present or not (Fig. 2a, b). We attribute this to the larger growth rate difference between strains AS and $AR_{C,Str}$ when compared to the growth rate difference between strains AS and $AR_{C,Tet}$ (i.e., for our experimental system, the fitness cost of streptomycin resistance is significantly greater than that of tetracycline resistance) (two-sample two-sided Welch test; $p = 8.2 \times 10^{-7}$, $n = 5$) (Fig. 2c and Supplementary Fig. 1). Thus, while phage lysis can increase the persistence of slower-growing AR strains in the face of faster-growing AS strains, this is potentially only true if the fitness cost of antibiotic resistance is not excessively large.

## Fitness cost of antibiotic resistance determines the persistence of AR cells

To test the notion that the effect of phage lysis on the persistence of slower-growing AR cells depends on the fitness cost of antibiotic resistance (i.e., that excessively large fitness costs can eliminate the positive effect of phage lysis), we adapted an individual-based computational model that allows us to simulate co-culture growth across nutrient-amended surfaces[40,44,45]. We performed simulations of co-cultures composed of a faster-growing AS strain (cyan) and a slower-growing AR strain (magenta) and varied the fitness cost of antibiotic resistance and the rate of phage lysis (Fig. 3a and Supplementary

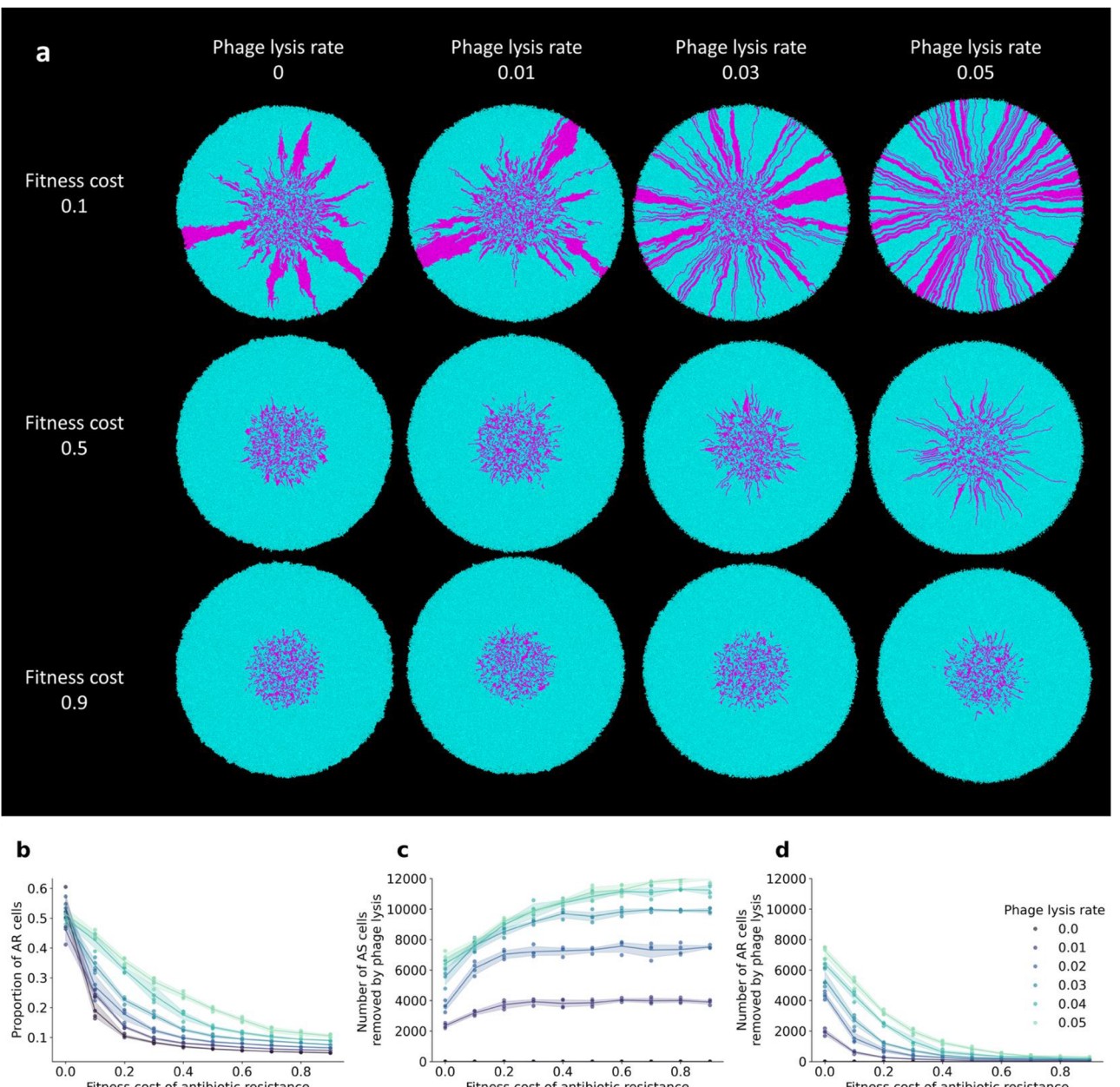

**Fig. 3 | Fitness cost and rate of phage lysis determine the persistence of slower-growing AR cells. a** Representative individual-based computational simulations of co-cultures of strains AS (cyan) and AR (magenta) in the absence or presence of phage lysis with different fitness costs of antibiotic resistance and rates of phage lysis. The images are the outputs at the last simulation time step. **b** The proportions of AR cells as a function of the fitness cost of antibiotic resistance for different rates of phage lysis. **c** The number of AS cells removed by phage lysis as a function of the fitness cost of antibiotic resistance for different rates of phage lysis. **d** The number of AR cells removed by phage lysis as a function of the fitness cost of antibiotic resistance for different rates of phage lysis. For **b**–**d**, each data point is an independent simulation ($n = 4$), the lines connect the mean values, and the shaded regions are ± one standard deviation from the mean. Source data are provided as a Source Data file.

Movie 1). As with our experiments, the faster-growing AS cells do not emerge spontaneously during the simulations; rather, they are already present in the inoculum. Also consistent with our experiments, we used a 1:1 initial cell ratio of the two strains and conducted the simulations in the absence of antibiotic pressure. We simulated phage lysis by removing peripheral cells at varying rates (between 0.01 and 0.05), which has been found to be a reasonable approximation of the effect of phage lysis[39,40].

We found that the effect of phage lysis on the persistence of slower-growing AR cells does indeed depend on the fitness cost of antibiotic resistance. When we set the rate of phage lysis to zero, the

AR strain had a clear growth disadvantage due to the fitness cost associated with antibiotic resistance (Fig. 3a), which is consistent with our experimental results (Fig. 2a). Overall, the proportion of AR cells at the final simulation time step decreased as the fitness cost of antibiotic resistance increased (Spearman rank correlation test; rho = −0.99, $p_{BC} = 8.1 \times 10^{-34}$) (Fig. 3b). When we then set the rate of phage lysis to a positive number, we observed increased persistence of the slower-growing AR cells (Pearson correlation test; rho = 0.94, $p_{BC} = 1.0 \times 10^{-11}$) (Fig. 3a, b). The effect size had a unimodal relationship with the fitness cost of antibiotic resistance, where the maximum beneficial effect of phage lysis on the persistence of AR cells occurred at a fitness cost of

antibiotic resistance of 0.1 (two-sample two-sided Welch tests; cost = 0.0 vs. cost = 0.1, $p_{BC} = 4.8 \times 10^{-4}$, $n = 4$; cost = 0.1 vs. cost = 0.9, $p_{BC} = 6.4 \times 10^{-3}$, $n = 4$) (Fig. 3b and Supplementary Fig. 3). The effect size of phage lysis then declined as the fitness cost of antibiotic resistance increased (Fig. 3b and Supplementary Fig. 3). Thus, the slower-growing AR cells persisted most effectively when the fitness cost of antibiotic resistance was relatively low (Fig. 3b and Supplementary Fig. 3), which is consistent with our experimental observations (Fig. 2a, b).

### Mechanism for how phage lysis increases the persistence of AR cells

To identify a plausible mechanism for how phage lysis can increase the persistence of slower-growing AR cells in the face of faster-growing AS counterparts, we again used our individual-based computational model to count the number of AR and AS cells that were removed from the biomass via phage lysis. We found that more AS cells than AR cells were removed via phage lysis when there was a fitness cost of antibiotic resistance (two-sample two-sided Welch tests; $p_{BC} = 4.6 \times 10^{-3}$, $n = 4$) (Fig. 3c, d). We refer to this outcome as peripheral kill-the-winner; faster-growing AS cells have a growth advantage that allows them to disproportionately occupy the biomass periphery, but cells at the biomass periphery are also more susceptible to removal via phage lysis. The disproportional removal of faster-growing AS cells diminishes the benefits of their faster growth rates, allowing slower-growing AR cells to persist. This concept is supported by our individual-based computational simulations, where we varied the rate of phage lysis. As the rate of phage lysis increased, the number of faster-growing AS cells that were removed from the biomass also increased (Spearman rank correlation test; rho = 0.90, $p_{BC} = 1.2 \times 10^{-8}$) (Fig. 3c), which correspondingly increased the persistence of slower-growing AR cells (Pearson correlation test; rho = 0.94, $p_{BC} = 1.0 \times 10^{-11}$) (Fig. 3d). While we analyzed all our simulations at a fixed simulation time for the data presented in Fig. 3, all our outcomes remain valid when we analyzed our simulations at a fixed total biomass size (Supplementary Fig. 4). Our main outcomes are therefore robust to the simulation endpoint.

To better understand the mechanisms driving the persistence of slower-growing AR cells, we tested whether differences in strain-specific phage infection traits can explain the persistence of slower-growing AR cells. To achieve this, we extended our individual-based computational simulations to systematically assess the effects of the phage lysis rate and the cost of antibiotic resistance for AR cells when the phage lysis rate for AS cells is fixed (Supplementary Fig. 5a). We found that modest differences in the phage lysis rate (e.g., AR cells being 0.01–0.02 less likely to be lysed that the AS cells) had only a minimal beneficial impact on AR persistence (Supplementary Fig. 5b). Under these conditions, AR cells persisted mainly due to their spatial positioning within the biomass interior that is less susceptible to phage lysis while AS cells dominated the periphery and were more susceptible to phage lysis. This is supported by the considerable AS cell lysis in these conditions, implying that AS cells still occupy the periphery that is exposed to phage attack (Supplementary Fig. 5c). Only when the phage lysis rates of the AS and AR cells were highly different (≥0.03) could the persistence of AR cells no longer be explained solely by the spatial protection of AS cells. Under these high-difference conditions, despite the high lysis rates on AS cells, almost no lysis of AS cells occurred, indicating that AS cells no longer occupied the periphery and were less exposed to phage attack (Supplementary Fig. 5c). These results indicate that while variations in phage lysis susceptibility may modulate ecological outcomes, it does not override the dominant influence of spatial structure and growth-driven spatial positioning. Overall, AR persistence emerges robustly from spatially biased phage predation, regardless of small differences in phage infection efficiency between strains.

### Phage lysis increases the persistence of slower-growing AR strains in the face of spontaneously emerging AS cells

We next tested whether our outcomes remain valid when antibiotic resistance is spontaneously lost during the experiment. To test this, we performed surface-associated growth experiments with strain $AR_{P,Chl}$, which contains the non-transmissible plasmid pEF001 that encodes for chloramphenicol resistance and green fluorescent protein (falsely colored magenta in our images). If plasmid pEF001 is lost from an $AR_{P,Chl}$ cell during cell division, the cell reverts to a non-fluorescent and faster-growing version that we refer to as an AS cell (uncolored). We confirmed that the non-fluorescent cells had indeed lost the resistance-bearing plasmid pEF001 rather than only the expression of the fluorescent protein-encoding gene by verifying that they could no longer grow on agar plates amended with chloramphenicol. We can therefore propagate the slower-growing $AR_{P,Chl}$ cells alone on nutrient-amended agar surfaces, track the spontaneous emergence and proliferation of faster-growing AS cells with CLSM, and quantify the persistence of $AR_{P,Chl}$ cells in the face of the newly formed AS cells.

We found that phage lysis can indeed increase the persistence of slower-growing AR cells in the face of spontaneously formed and faster-growing AS cells. In the absence of phage T6, we observed a substantial loss of plasmid pEF001 within the biomass (Fig. 4a). This corresponded to the proliferation of faster-growing AS cells and a decline in the proportion of slower-growing $AR_{P,Chl}$ cells from 100% in the initial inoculum to only 67% of the total biomass after 10 days of incubation (Fig. 4b). Thus, the plasmid-free faster-growing AS cells had a clear competitive advantage over the slower-growing $AR_{P,Chl}$ cells. When we then added phage T6 to the co-cultures, the proportion of slower-growing $AR_{P,Chl}$ cells declined by a significantly smaller extent from 100% in the initial inoculum to 94% after ten days of incubation (two-sample two-sided Welch test; $p = 3.6 \times 10^{-4}$, $n = 5$) (Fig. 4a, b). Thus, the presence of phage T6 reduced the ability of the faster-growing AS cells to establish over the slower-growing $AR_{P,Chl}$ cells. This was true even though the total biomass area declined (two-sample two-sided Welch test; $p = 6.1 \times 10^{-8}$, $n = 5$) (Fig. 4c). Finally, $AR_{P,Chl}$ cells were more abundant along the biomass periphery when phage T6 was present (two-sample two-sided Welch tests; $p < 1.5 \times 10^{-3}$, $n = 5$) (Fig. 4d). Thus, phage T6 allowed the slower-growing $AR_{P,Chl}$ cells to better occupy the biomass periphery where resources were plentiful, which is consistent with our proposed peripheral kill-the-winner dynamic (Fig. 1). These outcomes remained valid across various environmental conditions, including anoxic environments and different incubation temperatures (Supplementary Fig. 6).

### Mechanism for how phage lysis can increase the persistence of AR cells in the face of spontaneously emerging AS cells

To identify a plausible mechanism for how phage lysis can increase the persistence of AR cells despite the spontaneous emergence of faster-growing AS cells, we incorporated the spontaneous loss of antibiotic resistance into our individual-based computational model. The loss of antibiotic resistance in our model is generic and could occur via a genetic mutation or, as in our experiments, by plasmid loss. We then counted the number of AS cells that emerged from AR cells over the course of the simulations in the absence of antibiotic pressure. We set the entire initial population to be AR cells (magenta), each of which can spontaneously transform into a faster-growing AS cell (gray) according to a defined probability. We then varied the probability of losing antibiotic resistance to be between 0.01 and 0.03, and the fitness cost of antibiotic resistance to be a reduction in the growth rate between 5 and 30%. We simulated phage lysis as described in our simulations for chromosomal antibiotic resistance genetic determinants.

When we analyzed our simulations at a fixed simulation time, we found that phage lysis does indeed increase the persistence of slower-growing AR cells in the face of faster-growing AS cells. When we set the rate of phage lysis to zero, we observed extensive emergence and

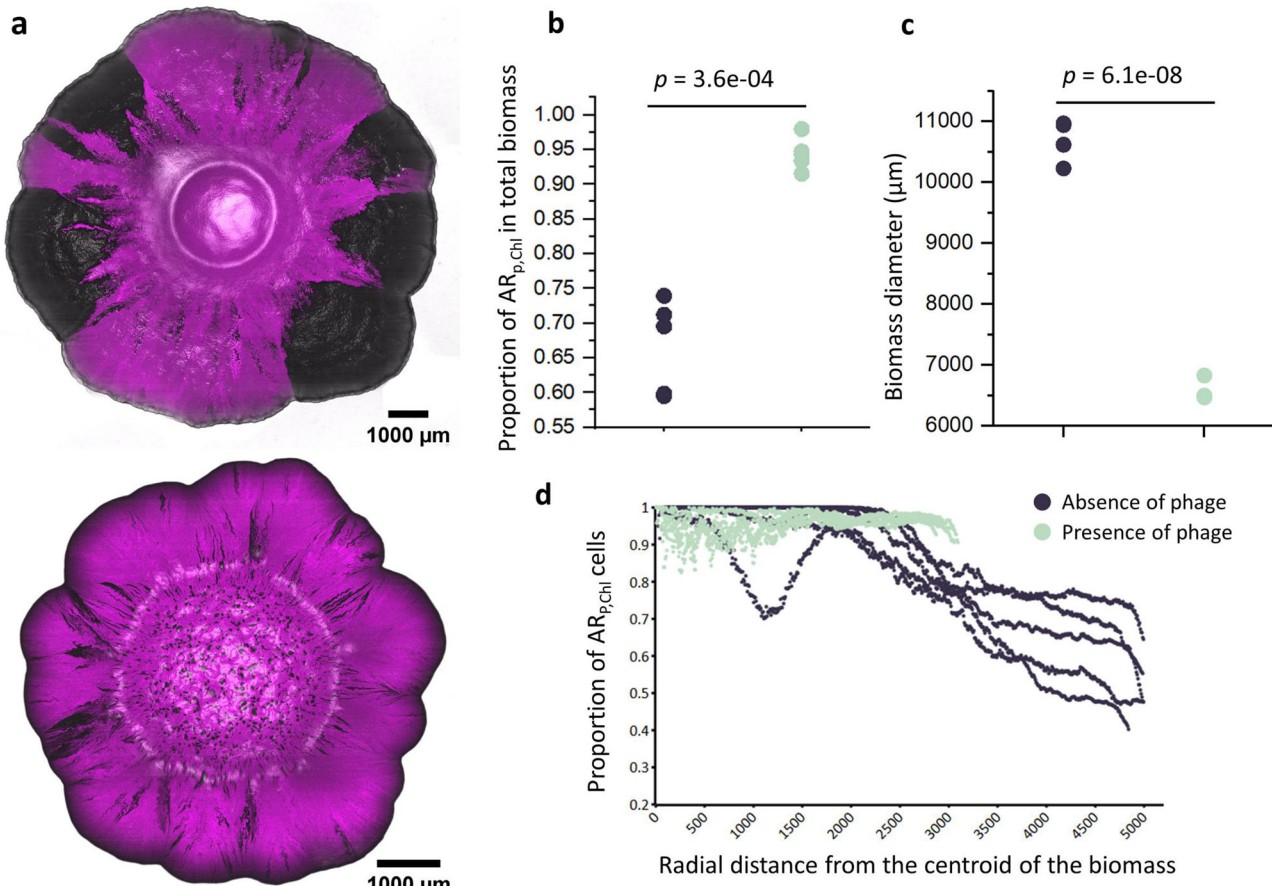

**Fig. 4 | Phage lysis increases the persistence of slower-growing AR cells in the face of spontaneously generated AS cells. a** Representative CLSM images of strain $AR_{P,Chl}$ (magenta) in the absence (upper image) or presence (lower image) of phage T6. Non-fluorescent regions are composed of $AR_{P,Chl}$ cells that spontaneously lost plasmid pEF001 and became AS cells. We took the images after 10 days of incubation at 21 °C in the absence of antibiotic pressure. **b** The proportions of the total biomass area occupied by strain $AR_{P,Chl}$ in the absence or presence of phage T6. **c** The biomass diameters in the absence or presence of phage T6. **d** The proportions of $AR_{P,Chl}$ cells as a function of the radial distance from the centroid of the biomass. For **b**–**d**, each data point is an independent experimental replicate ($n = 5$), the black data points are for experiments in the absence of phage T6, and the green data points are for experiments in the presence of phage T6. For **b**, **c**, the $p$ values are for two-sample two-sided Welch tests. Source data are provided as a Source Data file.

proliferation of faster-growing AS cells (Fig. 5a, c and Supplementary Movie 2), which is consistent with our experimental data (Fig. 4). The proportion of slower-growing AR cells significantly decreased as either the probability of losing antibiotic resistance increased (Spearman rank correlation test; rho = −0.72, $p_{BC}$ = 2.6 × 10$^{-3}$) or the fitness cost of antibiotic resistance increased (Spearman rank correlation test; rho = −0.90, $p_{BC}$ = 7.1 × 10$^{-10}$) (Fig. 5c). When we then set the rate of phage lysis to a positive number, the proportion of slower-growing AR cells significantly increased, particularly when the probability of losing antibiotic resistance was >0.02 and the fitness cost was >20% (two-sample two-sided Welch test; $p_{BC}$ = 4.1 × 10$^{-2}$, $n$ = 4) (Fig. 5b, c and Supplementary Movie 2). These outcomes remained valid when we analyzed our simulations at a fixed total biomass size (Supplementary Fig. 7), and they are therefore robust to the simulation endpoint.

To better understand the mechanism causing the increase in AR cells in the presence of phage, we quantified the proportions of slower-growing AR and faster-growing AS cells that are lysed by phage in our simulations. In the presence of phage, a larger proportion of the faster-growing AS cells were lysed compared to the slower-growing AR cells, especially when the AR cells carried a fitness cost and had a higher probability of losing antibiotic resistance (two-sample two-sided Welch tests; $p_{BC}$ = 2.4 × 10$^{-2}$, $n$ = 4) (Supplementary Fig. 8a). The increase in AR cells in the presence of phage cannot be explained by the number of AR cells that lost antibiotic resistance, as the number of

such events was either not substantially affected by or was higher in the presence of phage at the end of the simulations (two-sample two-sided Welch test; $p_{BC}$ = 4.7 × 10$^{-3}$, $n$ = 4) (Supplementary Fig. 9a). Thus, our proposed peripheral kill-the-winner dynamic remains valid even when AS cells emerge spontaneously. These outcomes remained valid when we analyzed our simulations at a fixed total biomass size (Supplementary Figs. 8b, 9b).

We further found that the properties of the spontaneously emerging AS cells are important determinants of the persistence of slower-growing AR cells. The number of AS cells that were lysed by phage significantly increased as the probability of losing antibiotic resistance increased (Spearman rank correlation test; rho = 0.88, $p_{BC}$ = 1.7 × 10$^{-6}$) but was relatively invariant across different fitness costs of antibiotic resistance (Supplementary Fig. 10). For a probability of losing antibiotic resistance of 0.1 and a fitness cost of 30%, over 2000 AS cells were removed from the biomass by phage lysis, underscoring the significant impact of phage on the faster-growing AS cells. The relationship between the number of AS cells and the probability of losing antibiotic resistance remained valid at a fixed biomass size, while the number of AS cells lysed by the phage increased as the fitness cost of antibiotic resistance increased (Spearman rank correlation test; rho = 0.97, $p_{BC}$ = 7.1 × 10$^{-19}$) (Supplementary Fig. 11a). Meanwhile, the number of slower-growing AR cells lysed by phage decreased as the probability of losing antibiotic resistance increased

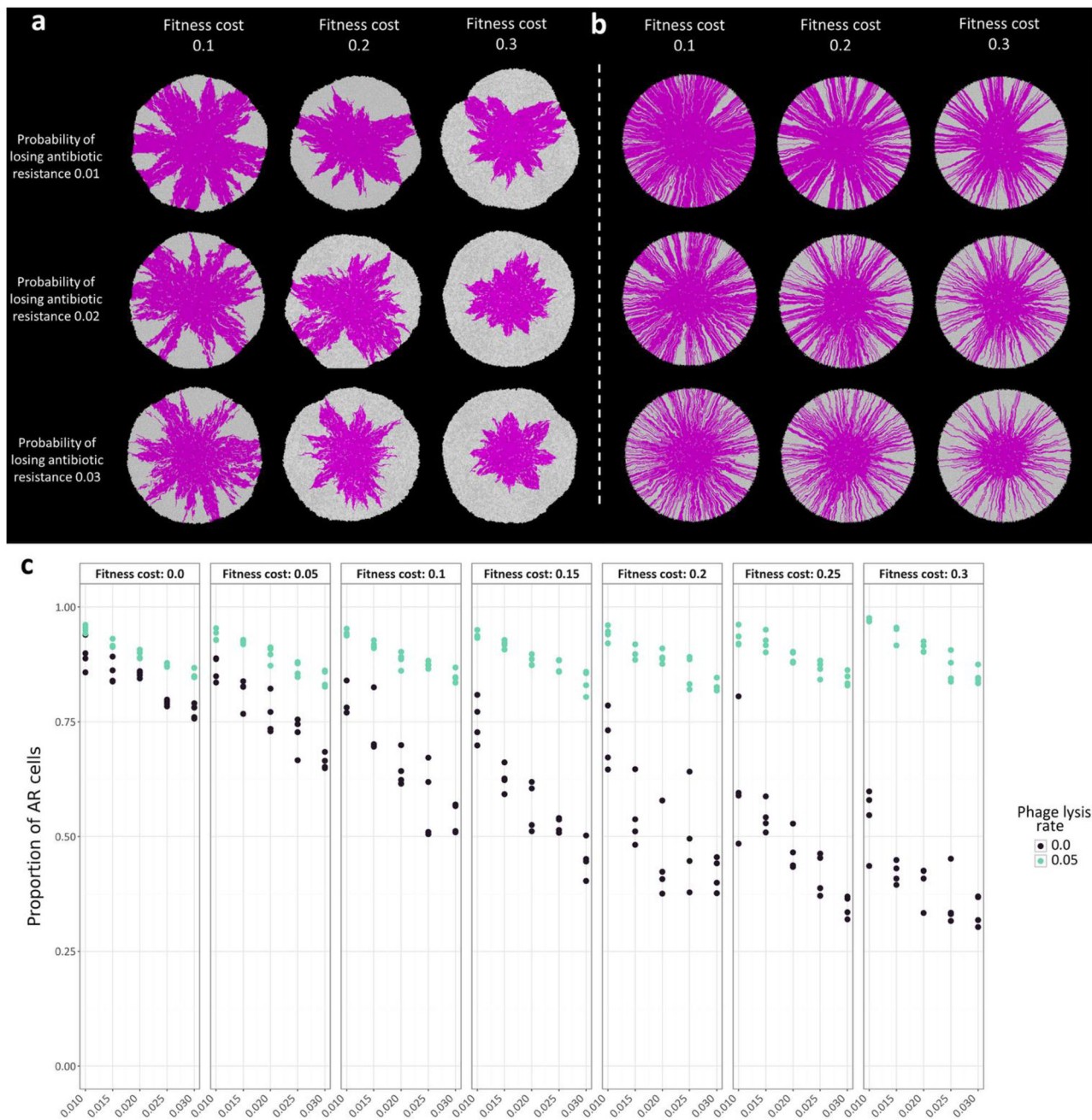

**Fig. 5 | Properties of spontaneously emerging AS cells determine the persistence of slower-growing AR cells. a, b** Representative individual-based computational simulations of strain AR (magenta) in the **a** absence or **b** presence of phage lysis with different fitness costs of antibiotic resistance and probabilities of losing antibiotic resistance. If AR cells undergo genetic changes that cause them to lose antibiotic resistance, such as the segregational loss of a plasmid, they are relieved of the fitness cost and become AS cells (gray). The images are the outputs at the last simulation time step. **c** The proportions of AR cells as a function of the fitness cost of antibiotic resistance and the probability of losing antibiotic resistance for different rates of phage lysis. Each data point is an independent simulation ($n = 4$), the black data points are in the absence of phage, and the green data points are in the presence of phage. Source data are provided as a Source Data file.

(Spearman rank correlation test; rho = −0.66, $p_{BC}$ = 1.2 × 10⁻²) or the fitness cost of antibiotic resistance increased (Spearman rank correlation test; rho = −0.99, $p_{BC}$ = 5.3 × 10⁻²²) (Supplementary Fig. 10). These outcomes remained valid when we analyzed our simulations at a fixed total biomass size (Supplementary Fig. 11b).

### Phage lysis promotes biodiversity during surface-associated growth
We finally sought to test whether our main outcomes can be generalized beyond simple binary co-cultures of two microbial strains. Can

our proposed peripheral kill-the-winner dynamic enable larger collections of microbial strains to persist together despite variance in their growth rates? To test this, we extended our individual-based computational model to simulate consortia consisting of between two and ten strains with different growth rates. We set the growth rate of one strain to a value of 1 and then randomly assigned the other strains to have growth rates ranging between 0 and 1, where we took each growth rate from a uniform distribution. In total, we performed 473 total simulations with 35 to 65 replicates for each number of strains in the consortia. We performed simulations in the absence or presence of

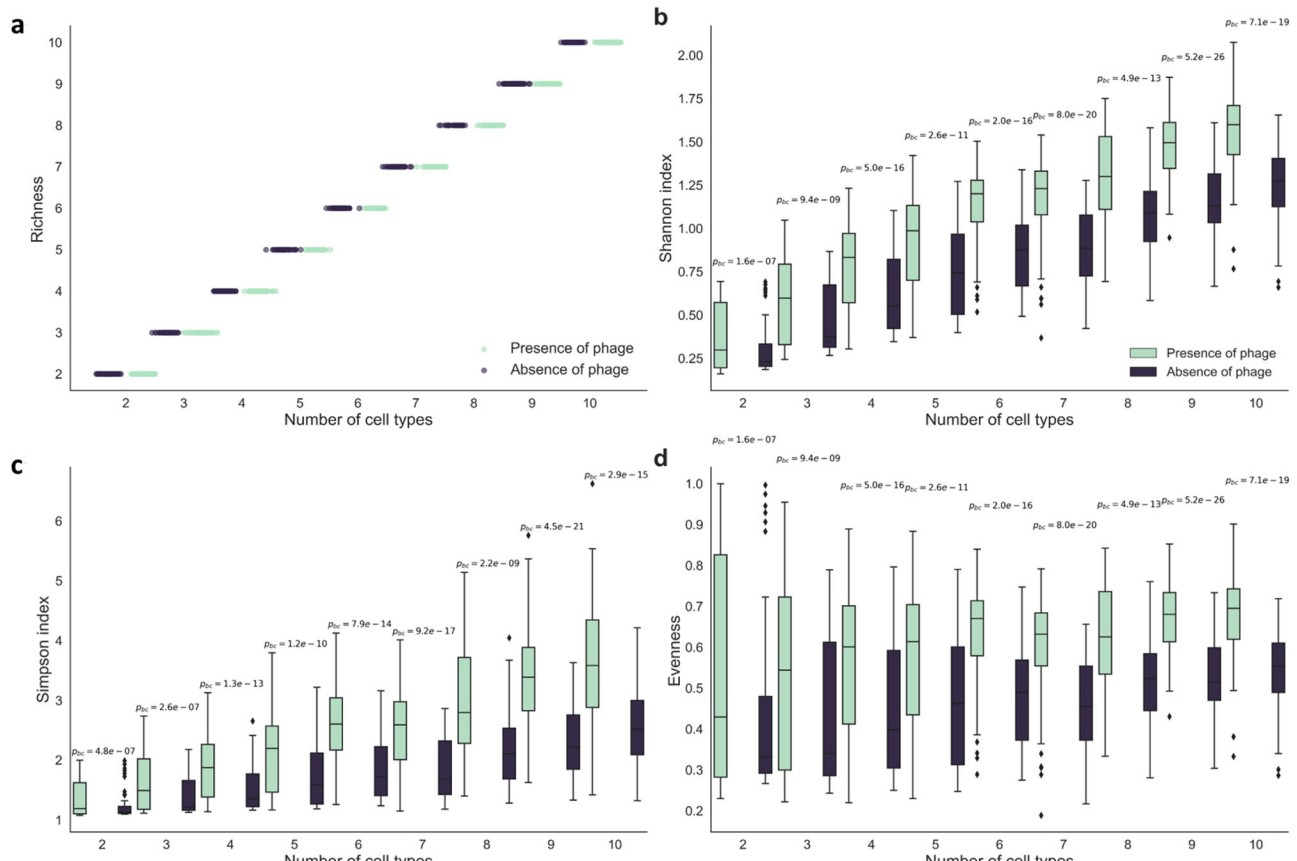

**Fig. 6 | Phage lysis maintains strain diversity.** Data were for individual-based computational simulations of the surface-associated growth of co-cultures consisting of between two and ten strains, where each strain has a different growth rate ranging between zero and one. The growth rate of one strain is one and the growth rates of the other strains are sampled from a uniform distribution. **a**–**d** The strain diversity metrics are for a fixed total biomass size and include **a** strain richness,

**b** Shannon diversity, **c** Simpson diversity, and **d** evenness. For **b**–**d**, the boxplots identify the mean values, interquartile ranges, and outliers for independent pairs of simulations ($n = 35$ for each number of cell types). The black boxplots and data points are in the absence of phage and the green boxplots and data points are in the presence of phage. The $p$ values are for two-sided paired $t$-tests with a Bonferroni correction. Source data are provided as a Source Data file.

phage as described above and quantified strain diversity from the final spatial patterns, including strain richness, Shannon diversity, Simpson diversity, and evenness.

We found that our proposed peripheral kill-the-winner dynamic can indeed be generalized to systems with more than two strains. The presence of phage did not impact strain richness (Fig. 6a), which is expected as we did not impose a mechanism of cell death other than phage lysis. Thus, all strains remained within the inoculation area, which is spatially protected from phage lysis. However, we found that the Shannon diversity (Paired two-sided $t$-test; $p_{BC} = 1.6 \times 10^{-7}$, $n > 35$) (Fig. 6b), Simpson diversity (Fig. 6c) (paired two-sided $t$-test; $p_{BC} = 4.8 \times 10^{-7}$, $n > 35$), and evenness (paired two-sided $t$-test; $p_{BC} = 1.6 \times 10^{-07}$, $n > 35$) (Fig. 6d) all significantly increased in the presence of phage. Thus, phage lysis not only preserved multiple strains with different growth rates (e.g., fitness costs of antibiotic resistance) but also contributed to a more even distribution of those strains (Fig. 6b–d and Supplementary Fig. 12). These outcomes remained valid when we analyzed our simulations at a fixed simulation time (Supplementary Fig. 13). Our proposed peripheral kill-the-winner dynamic may therefore be a general mechanism for how diversity can be maintained within surface-associated microbial systems.

## Discussion

Our findings demonstrate that phage lysis can reshape the spatial organization of surface-associated microbial systems with consequences on competitive outcomes between different

microorganisms. More specifically, we demonstrate a mechanism for how phage can enable the persistence of AR bacteria even in the absence of antibiotic pressure and when antibiotic resistance imposes a fitness cost. Consistent with prior studies that highlight the fitness costs of antibiotic resistance in the absence of antibiotic pressure[41,42], our experiments confirm that slower-growing AR strains are outcompeted by faster-growing AS strains when phage are absent. However, the presence of phage initiates a peripheral kill-the-winner dynamic where phage disproportionately lyse the faster-growing AS cells located at the biomass periphery (Figs. 2, 4). This creates ecological niches located behind the biomass periphery that allow slower-growing AR strains to persist despite their slower growth. These findings underscore the pivotal role of spatial organization in directing microbial community dynamics and evolution and reveal phage as important drivers of these processes. Importantly, these insights highlight the dual role of phage. On the one hand, they offer therapeutic potential against disease-causing or - exacerbating microorganisms[46–50]; on the other hand, their selective pressures on microbial systems may inadvertently maintain or even promote antibiotic resistance under certain conditions[40]. This duality necessitates careful consideration of the ecological consequences of phage therapy, particularly in scenarios where complete eradication of target populations is challenging.

A key insight from our study is the differential impact of phage lysis on AR strains with varying fitness costs. While a tetracycline-resistant strain ($AR_{C,Tet}$) substantially benefited from phage-mediated

ecological opportunities, the streptomycin-resistant strain ($AR_{C,Str}$) did not (Fig. 2). This discrepancy likely reflects the higher fitness cost of streptomycin resistance in our experimental system, which prevents the $AR_{C,Str}$ strain from capitalizing on phage-mediated niche creation (Fig. 3). These findings emphasize that the persistence of AR strains under phage pressure is contingent on the specific fitness burden associated with resistance mechanisms. Future research should explore this relationship across a broader spectrum of resistance determinants to refine predictions of resistance persistence under varying phage-bacteria dynamics.

Our individual-based computational simulations further revealed that minor differences in phage lysis rates between the AR and AS strains do not qualitatively alter our main outcomes (Supplementary Fig. 5). That is, spatial positioning and nutrient-driven growth segregation remain the primary factors enabling AR persistence even when AR strains experience slightly lower phage lysis susceptibility. Only when there are large differences in phage lysis rates do we observe significant effects on AR survival. While such differences may fine-tune phage infection dynamics in well-mixed environments, their ecological impact on structured systems is context-dependent and secondary to spatial exposure. Overall, our results emphasize that spatial organization and the peripheral kill-the-winner dynamic are the dominant forces driving the maintenance of AR cells, rather than inherent strain-level differences in phage susceptibility.

We further extended the concept of phage-mediated persistence of antibiotic resistance to systems subjected to the spontaneous emergence of antibiotic sensitivity (i.e., spontaneous loss of antibiotic resistance and its associated fitness costs), which could occur via genetic mutation or plasmid loss. This emergence of antibiotic sensitivity is more likely to occur in rapidly dividing cells that are more prone to segregational plasmid loss or DNA replication errors, which are often positioned at the biomass periphery where nutrients are abundant. Once relieved of the fitness cost associated with antibiotic resistance, the faster-growing AS cells gain further access to the biomass periphery. However, here we demonstrate that our proposed peripheral kill-the-winner dynamic can counteract other peripheral processes occurring in the system, ultimately preserving the antibiotic-resistant population (Figs. 4, 5). Our mechanism is thus able to explain the persistence of antibiotic resistance even in the presence of evolutionary processes that modify antibiotic resistance landscapes.

Beyond the maintenance of antibiotic resistance, our peripheral kill-the-winner dynamic offers novel insights into the maintenance of metabolically costly and non-transmissible plasmids. While conventional models attribute the persistence of such plasmids to horizontal gene transfer (HGT)[20,21,51,52], which can facilitate their dissemination and maintenance within microbial communities even though they impose a fitness cost, our findings demonstrate that HGT is not an obligatory mechanism for maintaining these plasmids. Instead, our results reveal an alternative ecological mechanism underpinning plasmid persistence driven by phage lysis. We show that phage preferentially lyse plasmid-free cells that are relieved of their fitness burden, thereby mitigating the competitive disadvantage typically associated with plasmid-bearing strains. This selective pressure not only enables plasmid-carrying antibiotic-resistant cells to persist but also enables them to increase in frequency in the presence of phage independent of HGT. While previous studies have established the role of phage in facilitating HGT[40,53,54], our findings underscore that plasmid maintenance can occur through phage-mediated selection even in the absence of HGT. This mechanism provides a compelling explanation for the evolutionary persistence of metabolically costly plasmids in both natural and clinical environments, expanding our understanding of the ecological forces shaping microbial genome evolution.

Our results are not specific to simple binary systems consisting of antibiotic resistant and sensitive counterparts but instead can be generalized to any number of strains that vary in their growth properties (Fig. 6). By selectively removing the fastest growing strains from the biomass periphery, phage will prevent any one strain from dominating the system and lead to a more balanced strain distribution. Our proposed peripheral kill-the-winner dynamic may therefore be relevant for understanding how many natural environments, such as soils and the human microbiome, are able to sustain such incredible levels of biodiversity. These habitats contain surface-associated microbial communities and can be abundant in phage[55–60], and it is therefore plausible that our peripheral kill-the-winner contributes to the maintenance of diversity within these systems.

While our study focuses on surface-associated bacterial colonies, it is important to consider how these dynamics might extend to biofilm-forming communities, which are prevalent in natural and clinical settings. Biofilms exhibit distinct structural and physiological properties that could modulate phage-host interactions. The matrix of extracellular polymeric substances (EPS) in biofilms can impede phage diffusion, potentially restricting access to bacterial populations at the periphery and altering the strength of peripheral kill-the-winner selection. Additionally, biofilm-associated heterogeneity, including the presence of metabolically inactive persister cells, could further influence the persistence of antibiotic-resistant subpopulations[61–63]. Future studies incorporating well-characterized biofilm models and assessing the impact of EPS composition and biofilm maturation on phage predation dynamics are important to fully understand the ecological significance of this mechanism in biofilm systems.

The reductionist approach we employed in this study provides a powerful means to identify and isolate mechanisms, but is limited in its generalizability. Our study is based on a simplified system involving a single bacterial species and a single phage, where the phage was introduced at a single point in time. While this approach enabled us to dissect the mechanism in detail by minimizing confounding factors, it does not capture the complexity of natural environments. In nature, phage-host dynamics can vary widely depending on host genetic background, phage infection strategies, and the presence of microbial competitors. Moreover, the timing of phage addition likely plays an important role in determining how phages modulate the spatial structure of biofilms, especially as diffusion limitations become more pronounced with increasing biomass. Future research should explore the extent of peripheral kill-the-winner effects across diverse bacterial and phage systems, in more complex environments, and within multispecies microbial communities to fully assess the ecological and evolutionary significance of this mechanism.

In conclusion, our study uncovers a mechanism that can help to explain the paradox of how antibiotic-resistant bacteria can persist in the absence of antibiotic pressure. By disproportionately lysing cells positioned at the biomass periphery, phage enable the persistence of slower-growing antibiotic-resistant strains, challenging traditional models of resistance loss in antibiotic-free conditions. These findings provide new insights into the ecological and evolutionary mechanisms underlying the spread of antibiotic resistance and highlight the complex interplay between phage, microbial spatial organization, and the maintenance of antibiotic resistance. Our results have implications for microbial management, phage therapy, and the global fight against antibiotic resistance, underscoring the importance of integrating spatial and ecological perspectives into resistance mitigation strategies.

## Methods

### Strains and culture conditions

We used isogenic derivatives of *E. coli* MG1655 for all our experiments. We used strain TB204 as the antibiotic-sensitive strain (referred to as strain AS), which expresses the *gfp* green fluorescent protein-encoding gene from the chromosome and is under the control of the lambda promoter[64]. We used strain TB205 to create all the antibiotic-resistant derivative strains used in this study, which is identical to strain TB204

except that it expresses the *mcherry* fluorescent protein-encoding gene from the chromosome and is under the control of the lambda promoter[64]. We created tetracycline and streptomycin resistant derivatives of strain TB205 (referred to as strains $AR_{C,Tet}$ and $AR_{C,Str}$, respectively) by growing cultures of strain TB205 in liquid lysogeny broth (LB) medium into stationary phase and then inoculating the cultures onto agar plates amended with increasingly large concentrations of tetracycline or streptomycin. We created a chloramphenicol-resistant derivative of strain *E. coli* MG1655 (referred to as strain $AR_{P,Chl}$) by introducing the non-mobile plasmid pEF001 into the strain via conjugation. This plasmid encodes for the *gfp* green fluorescent protein-encoding gene and chloramphenicol resistance (GenBank accession number PV021963). We routinely grew all the strains in liquid LB medium or on solid LB agar plates supplemented with $10 \, \mu g \, mL^{-1}$ tetracycline, $50 \, \mu g \, mL^{-1}$ streptomycin, or $25 \, \mu g \, mL^{-1}$ chloramphenicol, respectively, to maintain their resistance determinants. We preserved all strains in 15% (v/v) glycerol at −80 °C. Prior to each experiment, we streaked the individual strains from their respective −80 °C glycerol stocks onto LB agar plates containing their respective antibiotic and used a single colony to initiate all our experiments.

We used the lytic phage T6[65] for all our experiments. We propagated the phage using strain TB204 as the host. Briefly, we grew strain TB204 in LB liquid medium at 37 °C for 4 h with continuous shaking at 150 rpm. After incubation, we purified phage by filtering the culture through a 0.22 μm membrane and storing the supernatant at 4 °C until further use. For long-term storage, we mixed equal volumes of strain TB204 and phage suspensions and incubated the mixtures for 10 min with continuous shaking at 150 rpm. We then added glycerol to the mixtures (15% v/v) and stored the stocks at −80 °C.

### Surface-associated growth experiments
We performed surface-associated growth experiments across LB agar plates[40]. Briefly, we prepared LB agar plates by pouring autoclaved LB medium containing 1% bacteriology-grade agar into sterile 3.5 cm diameter Petri dishes. We then allowed the medium to solidify overnight at room temperature. We next transferred the agar plates to a sterile hood and allowed them to dry with their lids open for ten minutes. Finally, we covered the plates with their lids, sealed them individually with Parafilm (Amcor, Zürich, Switzerland), and stored them at 4 °C until use.

To perform the experiments, we first prepared overnight cultures of the individual strains in liquid LB medium containing the appropriate antibiotics to maintain their resistance determinants. We then diluted the overnight cultures 1:100 (v/v) into fresh LB medium in the absence of antibiotics and incubated them at 37 °C with continuous shaking at 150 rpm for 4 h to ensure that they were in the exponential growth phase. We next washed the cells to remove all traces of antibiotics by centrifugation at 3600×*g* for 10 min at 4 °C and resuspension in phosphate-buffered saline (PBS). Finally, we adjusted the optical density at 600 nm ($OD_{600}$) of each culture to 1 ($\sim 10^8$ colony-forming units $ml^{-1}$).

To prepare the phage for the experiments, we mixed the refrigerated phage stock with *E. coli* host cells and incubated the mixtures at 37 °C with continuous shaking at 150 rpm for 4 h. After incubation, we removed the host cells by filtration through a 0.22 μm membrane, resulting in a cell-free active phage solution. We determined the phage titer using the double-layer agar plate method[66] and then diluted the phage solution in PBS to obtain a concentration of $10^8$ plaque-forming units $ml^{-1}$.

For the surface-associated growth experiments with chromosomal antibiotic resistance genetic determinants, we placed a 1 μL droplet of a 1:1 mixture of strains AS and $AR_{C,Tet}$ or strains AS and $AR_{C,Str}$ ($OD_{600}$ of the mixture of 1) onto the centers of individual replicated LB agar plates. For surface-associated growth experiments with plasmid-encoded antibiotic resistance genetic determinants, we placed a 1 μl

droplet of strain $AR_{P,Chl}$ onto the centers of individual replicated LB agar plates. We then incubated the agar plates for 6 h. Thereafter, we added a 1 μL droplet of either phage solution or phage-free PBS as a control to the biomass of each agar plate. Finally, we incubated the agar plates in oxic conditions at 21 °C for ten days. We performed ten experimental replicates for each treatment. We then randomly selected five replicates that showed no detectable phage-resistant mutants for imaging and quantitative analysis. This allowed us to exclude the confounding effects of phage resistance evolution, which typically emerges in one to three replicates per treatment under our experimental conditions[40].

### Confocal laser scanning microscopy and image analysis
At the end of the surface-associated growth experiments, we imaged the biomass using a Leica TCS SP5 II confocal laser scanning microscope (CLSM) (Leica Microsystems, Wetzlar, Germany) equipped with a 2.5x HCX FL objective, a numerical aperture of 0.12, and a frame size of 512 × 512 pixels. We set the laser to 488 nm for the excitation of GFP fluorescence and 514 nm for the excitation of RFP fluorescence.

We quantified the proportions of different strains as a function of biomass radius using ImageJ (https://imagej.nih.gov/ij/) with Fiji plugins (v. 2.1.0/1.53c) (https://fiji.sc). We first loaded the image files into ImageJ and separated the channels to enable the analysis of each strain. We then selected a region of interest and applied an automated threshold using the Yen algorithm optimized for dark backgrounds. We next converted the images into binary masks to minimize background noise and enhance the signal-to-noise ratio. After isolating the target features, we applied a circular Region of Interest (ROI) of specified coordinates and radius to capture the spatial distributions of the strains within a consistent area. We executed the Radial Profile function to obtain the intensity distribution across this circular ROI. This distribution served as the basis for determining the radial intensity profile, which is essential for quantifying the community proportions along the circumference. Using the radial intensity profiles of each strain obtained from the ROI, we measured the relative abundances or proportions of each strain along the defined circumference. We repeated this process across all channels to ensure uniformity and comparability in data acquisition.

### Individual-based computational modeling
We employed the open-source modeling software Cellmodeller version 4.3 framework[44,45] (https://github.com/cellmodeller/CellModeller) and made modifications based on the MicrobialEcologyToolbox branch to carry out individual-based simulations of surface-associated microbial growth. We performed all simulations on the Euler computing cluster at ETH using the Slurm workload manager for batch job submissions. We wrote sections of the code for scheduling batch jobs with the aid of ChatGPT-4 (https://openai.com/gpt-4). We simulated bacterial cells as three-dimensional capsules of length L that grow uniaxially. Upon reaching a predetermined target length, cells are divided into two daughter cells, each inheriting the characteristics of the parent cell. We set the biophysical parameter gamma, which controls the ratio of drag force on cell growth, to 20 for all our simulations.

For the simulation of *E. coli* cells, we set the initial cell shape to have a radius of 0.5 μm and a length of 2 μm. Cells divided when their length reached a target length ($L_{div}$), where $L_{div}$ was randomly sampled from a Gaussian distribution of 2 to 2.5 μm, and daughter cells had a length of $L_{div}/2$. We set the growth rate of AS cells to 1, indicating that each cell would ideally grow by 1 μm per unit time step in the absence of physical constraints. To examine the impact of phage lysis, we implemented the killflag feature in CellModeller, which dynamically removed cells from the simulations and updated the cell state list to retain only living cells[40]. Briefly, the model determined the outermost ring of cells in the simulation space and, using a random probability (kill_rate), flagged cells to stop growing and be removed from the simulation. We varied the kill_rate, emulating phage lysis, between 0

(no phage lysis) and 0.5 (strong phage lysis) in our simulations, where we refer to this value as the phage lysis rate. Thus, instead of simulating the complex biophysical processes of phage replication and lysis, we modeled the phage as a signal that initiates cell removal[40].

Using our model, we set the initial conditions to achieve three objectives regarding the effect of phage lysis using the parameters presented in Supplementary Table 1. First, to model the growth of $AR_C$ cells, we initialized the simulations with 1000 cells each of AS and AR cells (1:1 ratio) randomly distributed and rotationally oriented across a circular area of radius 50. We set the growth rate of the AR cells to be between 0.1 and 1, which corresponds to a fitness cost of antibiotic resistance between 0.9 and 0. We performed simulations until the total number of cells reached 40,000. Second, to model the growth of $AR_P$ cells, we initialized the simulations with 2000 $AR_P$ cells with a fitness cost for carrying the plasmid. We simulated the spontaneous emergence of AS cells through plasmid loss during cell division by modifying the division function of cells. Briefly, with a certain probability during cell division, $AR_P$ cells can switch to AS cells that no longer carry a fitness cost. We varied the fitness cost of antibiotic resistance to be between 0 and 0.3 and the probability that an AR cell will turn into an AS cell to be between 0.01 and 0.03. We performed simulations until the total number of cells reached 40,000. Third, to model the growth of multiple strains that differ in their growth rate, we initialized the simulations with 2000 total cells, comprised of equal numbers of between two and ten strains. We next set the fitness cost of the first strain to 0 and randomly sampled from a uniform distribution between 0 and 1 to assign fitness costs to all the other strains in the simulation. Thus, the growth rate of all the other strains ranged between 1 and 0 (fitness cost between 0 and 1) relative to the first strain. We performed simulations until the total number of cells reached 40,000 and then quantified the strain richness, Shannon diversity, Simpson diversity, and evenness using standard methods[67].

### Statistical analyses

We performed all statistical tests with Python (version 3.11.5) using the scipy.stats module (version 1.11.4). To test for differences between means, we employed two-sample two-sided Welch tests for comparing independent groups, which do not assume homoscedasticity in the datasets, and paired $t$-tests when comparing the same samples under two different treatments. For correlation measurements, we used Pearson's correlation test for linear relationships and Spearman's rank correlation test for monotonic but nonlinear relationships. When performing statistical tests on the same data multiple times, we adjusted the $p$ values using the Bonferroni correction, which we designate as $p_{BC}$. We tested the normality of our datasets using the Shapiro–Wilk test and did not observe significant deviations from the assumption of normality ($p < 0.05$). For each statistical test, we reported the specific test used, the corresponding $p$ value, and the sample size in the Results section.

### Reporting summary

Further information on research design is available in the Nature Portfolio Reporting Summary linked to this article.

## Data availability

All data generated in this study have been deposited in the publicly accessible Eawag Research Data Institutional Repository (https://opendata.eawag.ch/) at https://doi.org/10.25678/000EDH. Source data are provided with this paper.

## Code availability

All code generated in this study have been deposited in the publicly accessible Eawag Research Data Institutional Repository (https://opendata.eawag.ch/) at https://doi.org/10.25678/000EDH.

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

## Acknowledgements

We thank Zachary Bailey for providing phage T6; Dr. Martin Ackermann and Emanuele Fara for providing all *E. coli* strains and plasmid pEF001; and Drs. Mireia Cordero and Josep Ramoneda for helpful discussions. We thank Mrs. Ruan (Guo Chen) for her excellent assistance with structuring the Supplementary Movies. C.R. and D.P.V. were supported by a grant from the Swiss National Science Foundation (310030_207471) awarded to D.R.J.

## Author contributions

C.R. and D.P.V. conceived and developed the research question. C.R., D.P.V., and D.R.J. designed the experiments. C.R. performed all the experiments. D.P.V. performed all the individual-based computational simulations. C.R. and D.P.V. analyzed the data. C.R. and D.P.V. wrote the first version of the manuscript with contributions from D.R.J. D.R.J. coordinated the project. All authors reviewed and approved the final version of the manuscript.

## Competing interests

The authors declare no competing interests.
