## [Transparent Peer Review file · Nature Communications]

Phage-mediated peripheral kill-the-winner facilitates the maintenance of costly antibiotic resistance

Corresponding Author: Dr David Johnson

Version 0:

Reviewer comments:

Reviewer #1

(Remarks to the Author)

The authors have shown the selective pressures exerted by phages on bacteria sensitive to antibiotics. Most antibiotics kill growing bacteria, but this study demonstrates that phages also lyse bacteria that are able to grow at a faster rate. The agar experiments are also more relevant as they simulate natural environments where bacteria grow in biofilms.

However, the study does not directly measure the fitness (growth rates) of antibiotic resistant and sensitive strains. Growth curves for each bacteria should be able to detect the difference in the growth rates. Also, experiments with phages will also be helpful in noting the differences in phage traits such as phage burst size or eclipse period between the two strains. Correlating these traits in liquid culture will provide more robust evidence of events occurring on solid surfaces, although diffusion is much lower in microcolonies compared to well-mixed liquid cultures.

(Remarks on code availability)

Reviewer #2

(Remarks to the Author)

Spontaneous occurrence and transfer of antibiotic resistances pose a huge threat for the treatment of bacterial infections. Spontaneous development of antibiotic resistances by mutations or maintenance of resistance genes on plasmids or other MGEs usually comes with a significant metabolic burden, often leading to rapid loss of the trait in the absence of selective pressure. However, earlier studies showed that such mutants can persist for long periods of time in bacterial populations, and there are several hypotheses concerning the mechanisms underlying this persistence. In this manuscript, Ruan and co-workers developed a novel very intriguing hypothesis involving phage-host-interaction: By a so-called „peripheral kill-the-winner“ dynamic, faster growing cells are more rapidly removed by phage predation from the periphery of a growing colony, where fastest growth occurs. By this, the overall fraction of slower growing cells is stabilized or even increases. To validate this hypothesis, the authors conducted in vitro experiments on spontaneously resistant mutants or plasmid-harboring strains of *E. coli*. By fluorescent labeling in concert with CLSM, the authors followed and quantified the dynamics of a surface-grown colony over time. By an intricate modeling approach, they validated and expanded their model in silico. Taken together, the results indicate that phage predation in growing colonies can, in fact, stabilize the subpopulation of slowly growing cells, which does not necessarily relate only to antibiotic resistances but also mixed-species populations. However, this only occurs if the disadvantage in cell proliferation/growth is not too extensive.

Overall, the authors developed a highly intriguing hypothesis. While the modeling approach - as based on the variables that were considered – seems clear, there are, however, some issues I have with the in vivo approaches that need to be clarified.

- do the authors consider the occurrence of escape mutants that are not affected by phage predation anymore?

- is it clear that the mutant/plasmid bearing strains still have the same general susceptibility than the wild type? T6 requires the Tsx nucleoside permease for infection, is this an issue?

- How does T6 generally behave in slower growing strains, is the burst size affected?

- along these lines, how do WT & mutant strains behave in a non-mixed colony? Maybe AR C, Tet is generally more susceptible?

- Fig. 4a (and S3), here it seems that even in the presence of phage, the whole fringe of the colony has lost the fluorescence. Is that an optical/microscopical artifact? Is this an appearance of escape mutants? In the simulations, the colonies do not show this appearance.

- when the authors score for loss of resistance bearing plasmids: Is this truly the loss of the plasmid or is it just a loss of fluorescence? How about the growth rate of 'dark cells' isolated from the colonies?

- If I understood correctly, the phages are added quite early to the nascent colony – does this really only affect cells growing at the fringes of this colony?

- Fig. 2a, lower panel, in the absence of phage, some AR C, Str cells are retained, but seem to be completely absent in the presence of phages. How do the authors (and the simulations) explain this?

Minor:

- maybe use a more contrasting color for the phages in Fig.1

- 107 ff, this part I would move to the results part, it's not an introduction anymore.

- 484, space after 'burden'.

(Remarks on code availability)

Reviewer #3

(Remarks to the Author)

This paper presents a beautiful study of phage lysis on the edge of growing bacterial colonies and proves that this predation provides self-organized protection for the slower-growing bacteria. The study nicely combines experiment and modeling and is a delight to read.

My only comment concerns the formulation of the model. The authors define phage as a killing probability at the periphery of a colony. I guess the model was implemented as a killing rate, where bacteria on the periphery have a chance to be eliminated per unit timestep.

An even smaller comment concerns the somewhat boring but very professional title of the manuscript. I liked the "Peripheral kill the winner" concept emphasized throughout the manuscript, and maybe the authors could consider integrating this into a more eye-catching title.

I recommend the publication of this manuscript.

(Remarks on code availability)

Reviewer #4

(Remarks to the Author)

This study investigates how lytic phages can modulate bacterial spatial organization to facilitate the persistence of antibiotic-resistant bacteria, even in the absence of antibiotic pressure. Using a combination of experimental and computational approaches, the authors propose and test the "peripheral kill-the-winner" hypothesis, which suggests that phages disproportionately lyse faster-growing antibiotic-sensitive bacteria at the biomass periphery, enabling slower-growing AR bacteria to persist. These findings have implications for understanding antibiotic resistance maintenance, eco-evo dynamics, and the application of phage therapy in bacterial population control.

This is a well-executed and robust study, with strong experimental design and clear data presentation. However, my main concern is that the broader significance of this mechanism remains unclear. While the study convincingly demonstrates this effect under controlled experimental conditions, its relative importance compared to other mechanisms is not fully addressed. This uncertainty could limit the study's impact, particularly in real-world microbial communities where multiple competing factors influence these dynamics. My comments below are thus mostly in this vein—focusing on clarifying how significant this mechanism is in the broader context of antibiotic resistance maintenance and microbial ecology.

Comments

While the study discusses bacterial spatial organization, it does not explicitly address biofilms, which are crucial to understanding phage-bacteria interactions. Biofilms can provide protection against phage infection due to extracellular polymeric substances and structural heterogeneity. A discussion of how biofilm formation might alter the "peripheral kill-the-winner" mechanism would strengthen the ecological relevance of the study.

The paper convincingly demonstrates the effect of phage predation on AR persistence in specific experimental conditions. However, it is difficult to anticipate the relative importance of this mechanism compared to other well-documented resistance maintenance strategies, such as compensatory mutations, plasmid persistence, and co-selection with other traits. A discussion on whether this mechanism is likely to play a dominant or minor role in natural environments would be valuable. Similarly, the study uses only one bacterial strain and one phage, under highly controlled conditions. While this simplicity allows for clear conclusions, it raises concerns about generalizability.

The authors created resistant strains through experimental evolution. However, it is unclear whether multiple independent resistant strains were tested or if conclusions were drawn from a single resistant colony after serial passage. Clarifying what the biological replicates represent would help in assessing the robustness of the findings. If a different resistant colony would've been picked, would the conclusions be different?

(Remarks on code availability)

-

Reviewer #5

(Remarks to the Author)

(Remarks on code availability)

Version 1:

Reviewer comments:

Reviewer #1

(Remarks to the Author)

All the comments and concerns have been addressed in the revised version of the manuscript.

(Remarks on code availability)

Reviewer #2

(Remarks to the Author)

The authors have fully addressed my concerns, congratulations to that nice study!

(Remarks on code availability)

Reviewer #4

(Remarks to the Author)

The authors have adequately addressed my previous comments regarding the broader ecological significance of the peripheral kill-the-winner mechanism, the role of biofilm structure, the limitations of using a single strain-phage pair, and the clonality of resistant mutants. While some responses remain primarily textual and qualitative, I find them reasonable within the scope of this study. The additions to the Discussion section appropriately temper the authors' claims and clarify the boundaries of inference.

(Remarks on code availability)

none

Reviewer #5

(Remarks to the Author)

(Remarks on code availability)

Reviewer #1 (Remarks to the Author):

The authors have shown the selective pressures exerted by phages on bacteria sensitive to antibiotics. Most antibiotics kill growing bacteria, but this study demonstrates that phages also lyse bacteria that are able to grow at a faster rate. The agar experiments are also more relevant as they simulate natural environments where bacteria grow in biofilms. However, the study does not directly measure the fitness (growth rates) of antibiotic resistant and sensitive strains. Growth curves for each bacteria should be able to detect the difference in the growth rates. Also, experiments with phages will also be helpful in noting the differences in phage traits such as phage burst size or eclipse period between the two strains. Correlating these traits in liquid culture will provide more robust evidence of events occurring on solid surfaces, although diffusion is much lower in microcolonies compared to well-mixed liquid cultures.

We sincerely thank the reviewer for these constructive suggestions and for highlighting the importance of directly quantifying the fitness differences and potential phage trait variations between the AS and AR strains. We have now conducted liquid culture growth curve experiments to directly measure and compare the specific growth rates of the AS, AR_{C,Tet}, and AR_{C,Str} strains. As expected, our results confirm that both AR strains exhibit significantly reduced growth rates relative to the AS strain, with the AR_{C,Str} strain having the highest fitness cost. These quantitative measurements strengthen the conclusions drawn from our spatial assays on agar surfaces. We now provide this data in the Supplementary Information as a new extended data figure (Extended Data Fig. 1). We state the outcome of this experiment and reference the new supplementary figure in the following lines of the revised manuscript.

Lines 152-156: “We performed additional batch culture growth experiments to further corroborate these observations, confirming that both AR strains grow slower than strain AS, with strain AR_{C,Str} exhibiting the highest fitness cost (two-sample two-sided Welch tests; AS vs AR_{C,Tet}: $p = 1.6 \times 10^{-8}$; AS vs AR_{C,Str}: $p = 5.3 \times 10^{-9}$; AR_{C,Tet} vs AR_{C,Str}: $p = 6.5 \times 10^{-5}$) (Extended Data Fig. 1).”

Extended Data Fig. 1: Growth kinetics of strains AS and AR_{C,Tet} and AR_{C,Str} in batch culture. a, Optical density at 600 nm (OD₆₀₀) measurements over time for strains AS, AR_{C,Tet} and AR_{C,Str} at 21°C in the absence of phage. Each line connects the mean values and the shaded region is ± one standard deviation from the mean (n = 5). b, Maximum growth rates of each strain in the absence of phage. Each colored data point is an independent experimental replicate. The black data point is the mean value and the black vertical line is ± one standard deviation from the mean (n = 5). The p-values are for two-sample two-sided Welch tests with a Bonferroni correction.

We also performed phage infection assays in batch cultures to determine the burst size and latency period of phage T6 when infecting the AS and AR strains. Our data show that there are no significant differences between the burst size and latency period of the phage in the AR and AS strains. We now provide this data in the Supplementary Information as a new extended data figure (Extended Data Fig. 2). We state the outcome of this experiment and reference the new supplementary figure in the following lines of the revised manuscript.

Lines 185-188: “Furthermore, the burst sizes and latency periods of phage T6 also did not significantly differ among the AR strains, which provides further support that phage infection dynamics remained consistent despite the mutations (Extended Data Fig. 2).”

Extended Data Fig. 2: Phage one-step growth assays in batch culture. **a**, Numbers of plaque forming units (pfu/ml) estimated in an 80-minute period for strains AS, AR_{C,Tet} and AR_{C,Str}. Each line connects the mean values and error bars are \pm one standard deviation from the mean ($n = 5$). **b**, Burst sizes of phage for strains AS, AR_{C,Tet} and AR_{C,Str} determined from the phage growth curves. Each data point is an independent experimental replicate. The p -values are for two-sample two-sided Welch tests with a Bonferroni correction.

To further assess the effect of differences in phage infection traits for the two strains, we investigated the impact of reduced susceptibility to phage lysis of the of the slower-growing AR cells. To accomplish this, we repeated our individual-based simulations by varying the lysis rate of the AR cells between 0 to 0.05 while keeping the lysis rate of AS cell fixed at 0.05. We present these new data in Extended Data Fig. 5. Our simulations reveal that modest differences in the phage lysis rates between the two cell types (≤ 0.02) have only limited effects on AR persistence. In this regime, spatial structure and fitness-driven segregation remain the dominant determinants of community dynamics, with faster-growing AS cells disproportionately occupying the biomass periphery and being preferentially lysed. However, at lower AR cell phage lysis rates (e.g., ≤ 0.02), even small fitness costs exert a more pronounced effect on AR cell abundance. These results indicate that differences in phage infection traits can modulate but do not overturn the ecological outcome, underscoring the robustness of the "peripheral kill-the-winner" mechanism across a range of biologically plausible parameters. We state the outcome of this analysis referencing the following Extended Data figure in the results and have expanded the Discussion section to reinforce the robustness of our main conclusions.

Extended Data Fig. 5: Effect of the fitness cost of antibiotic resistance and the rate of phage lysis on the persistence of AR cells. **a**, Representative individual-based computational simulations of co-cultures of faster-growing AS cells (cyan) and slower-growing AR cells (magenta) for different combinations of the fitness cost and rate of phage lysis for AR cells. The rate of phage lysis for AS cells was fixed at 0.05. **b**, The final proportions of AR cells as a function of the fitness cost for different rates of phage lysis. **c**, The total numbers of AS cells removed by phage lysis as a function of the fitness cost and rate of phage lysis. **d**, The total numbers of AR cells removed by phage lysis. For b-d, each data point is an independent simulation ($n = 3$), the lines connect the mean values, and the shaded areas are \pm one standard deviation from the mean.

Lines 274-293: “To better understand the mechanisms driving the persistence of slower-growing AR cells, we tested whether differences in strain-specific phage infection traits can explain the persistence of slower-growing AR cells. To achieve this, we extended our individual-based computational simulations to systematically assess the effects of the phage lysis rate and the cost of antibiotic resistance for AR cells when the phage lysis rate for AS cells is fixed (Extended Data Fig. 5a). We found that modest differences in the phage lysis rate (e.g., AR cells being 0.01–0.02 less likely to be lysed than the AS cells) had only a

minimal beneficial impact on AR persistence (Extended Data Fig. 5b). Under these conditions, AR cells persisted mainly due to their spatial positioning within the biomass interior that is less susceptible to phage lysis while AS cells dominated the periphery and were more susceptible to phage lysis. This is supported by the considerable AS cell lysis in these conditions, implying that AS cells still occupy the periphery that is exposed to phage attack (Extended Data Fig. 5c). Only when the phage lysis rates of the AS and AR cells were highly different (≥ 0.03) could the persistence of AR cells no longer be explained solely by the spatial protection of AS cells. Under these high-difference conditions, despite the high lysis rates on AS cells, almost no lysis of AS cells occurred, indicating that AS cells no longer occupy the periphery and are less exposed to phage attack (Extended Data Fig. 5c). These results indicate that while variations in phage lysis susceptibility may modulate ecological outcomes, it does not override the dominant influence of spatial structure and growth-driven spatial positioning. Overall, AR persistence emerges robustly from spatially biased phage predation, regardless of small differences in phage infection efficiency between strains.”

Lines 490-499: *“Our individual-based computational simulations further revealed that minor differences in phage lysis rates between the AR and AS strains do not qualitatively alter our main outcomes (Extended Data Fig. 5). That is, spatial positioning and nutrient-driven growth segregation remain the primary factors enabling AR persistence even when AR strains experience slightly lower phage lysis susceptibility. Only when there are large differences in phage lysis rates do we observe significant effects on AR survival. While such differences may fine-tune phage infection dynamics in well-mixed environments, their ecological impact on structured systems is context-dependent and secondary to spatial exposure. Overall, our results emphasize that spatial organization and the “peripheral kill-the-winner” dynamic are the dominant forces driving the maintenance of AR cells, rather than inherent strain-level differences in phage susceptibility.”*

Reviewer #2 (Remarks to the Author):

Spontaneous occurrence and transfer of antibiotic resistances pose a huge threat for the treatment of bacterial infections. Spontaneous development of antibiotic resistances by mutations or maintenance of resistance genes on plasmids or other MGEs usually comes with a significant metabolic burden, often leading to rapid loss of the trait in the absence of selective pressure. However, earlier studies showed that such mutants can persist for long periods of time in bacterial populations, and there are several hypotheses concerning the mechanisms underlying this persistence. In this manuscript, Ruan and co-workers developed a novel very intriguing hypothesis involving phage-host-interaction: By a so-called “peripheral kill-the-winner” dynamic, faster growing cells are more rapidly removed by phage predation from the periphery of a growing colony, where fastest growth occurs. By this, the overall fraction of slower growing cells is stabilized or even increases. To validate this hypothesis, the authors conducted in vitro experiments on spontaneously resistant mutants or plasmid-harboring strains of *E. coli*. By fluorescent labeling in concert with CLSM, the authors followed and quantified the dynamics of a surface-grown colony over time. By an intricate modeling approach, they validated and expanded their model in silico. Taken together, the results indicate that phage predation in growing colonies can, in fact, stabilize the subpopulation of slowly growing cells, which does not necessarily relate only to antibiotic resistances but also mixed-species populations. However, this only occurs if the disadvantage in cell proliferation/growth is not too extensive. Overall, the authors developed a highly intriguing hypothesis. While the modeling approach - as based on the variables that were considered - seems clear, there are, however, some issues I have with the in vivo approaches that need to be clarified.

We sincerely thank the reviewer for their thorough evaluation of our study and their positive assessment of the novelty and significance of our proposed "peripheral kill-the-winner" hypothesis. Below, we address each specific point raised by the reviewer in detail and have revised the manuscript accordingly where appropriate.

- do the authors consider the occurrence of escape mutants that are not affected by phage predation anymore?

We thank the reviewer for raising this important point. We agree that the spontaneous emergence of phage-resistant escape mutants is a possible outcome during phage-bacteria interactions. To address this issue, we designed our experiments with sufficient biological replicates ($n = 10$ per treatment) to account for such stochastic events. In practice, we occasionally observed the emergence of phage-resistant escape mutants in 1 to 2 out of 10 replicates during the ten-day surface-associated growth experiments, which we reported and described previously in Ruan et al., 2024, *Nat Commun*. To ensure that our analyses accurately reflect the phage-mediated dynamics on susceptible bacterial populations, we only selected those replicates without detectable escape mutants for confocal laser-scanning microscopy (CLSM) imaging and quantitative analysis as described in Ruan et al., 2024, *Nat Commun*. We have now clarified this procedure in the Methods section of the revised manuscript.

Lines 648-653: *"We performed ten experimental replicates for each treatment. We then randomly selected five replicates that showed no detectable phage-resistant mutants for imaging and quantitative analysis as described elsewhere⁴⁰. This allowed us to exclude the confounding effects of phage resistance evolution, which typically emerges in one to three replicates per treatment under our experimental conditions."*

- is it clear that the mutant/plasmid bearing strains still have the same general susceptibility than the wild type? T6 requires the Tsx nucleoside permease for infection, is this an issue?

We thank the reviewer for raising this question. We carefully considered this issue in our experimental design. All AR strains used in this study, both chromosomal mutants ($AR_{C,Tet}$ and $AR_{C,Str}$) and the plasmid-bearing strain ($AR_{P,chl}$) were directly derived from the AS parental strain and are isogenic except for the specific resistance determinants. To empirically confirm that phage susceptibility remained comparable, we conducted phage spot assays and liquid infection assays with phage T6 on all strains prior to performing the surface-associated experiments. The monoculture liquid experiments in the presence of phage showed a decline in the cell density of all three strains after a certain period of time (see our response to Reviewer 1 and the new Extended Data Fig. 2b), confirming that all the AR strains have the same general susceptibility to the T6 phage despite the chromosomal mutations that they carry. In monoculture colonies, the diameter of all three strains decreased following the addition of phage (Fig. 2c), providing further evidence that the chromosomal mutant strains remained susceptible to phage lysis. A detailed explanation of the specific experimental data has been included in our response to Reviewer 1. We now state this for the reader in the following lines of the revised manuscript.

Lines 182-188: *"The increased persistence of the $AR_{C,Tet}$ cells could not be explained by differences in its susceptibility to phage T6. Despite their chromosomal mutations, both AR strains remained susceptible to phage T6, forming smaller biomass diameters on agar plates (Fig. 2c). Furthermore, the burst sizes and latency periods of phage T6 also did not significantly differ among the AR strains, which provides further support that phage infection dynamics remained consistent despite the mutations (Extended Data Fig. 2)."*

Fig. 2: Phage lysis increases the persistence of slower-growing AR cells. **a**, Representative CLSM images of co-cultures of strains AS (cyan) and AR_{C,Tet} (magenta) (upper images) or of strains AS (cyan) and AR_{C,Str} (magenta) (lower images) in the absence or presence of phage T6. We imaged the biomass after ten days of incubation in the absence of antibiotic pressure. **b**, The proportions of the total biomass areas occupied by strains AR_{C,Tet} or AR_{C,Str} when grown in co-culture with strain AS in the absence or presence of phage T6. **c**, The biomass diameters of strains AS, AR_{C,Tet} and AR_{C,Str} when grown in monoculture in the absence (left) or presence (right) of phage T6. **d**, The biomass diameters of co-cultures of strains AS and AR_{C,Tet} or strains AS and AR_{C,Str} in the absence or presence of phage T6. **e**, The biomass areas (population size) of strain AR_{C,Tet} when grown in co-culture with strain AS in the absence or presence of phage T6. **f**, The proportions of AR_{C,Tet} cells within co-cultures as a function of the radial distance from the centroid of the biomass. For **b–f**, each data point is an independent experimental replicate ($n = 5$), the black data points are for experiments in the absence of phage T6, and the green data points are for experiments in the presence of phage T6. For **b–e**, the p -values are for two-sample two-sided Welch tests.

- How does T6 generally behave in slower growing strains, is the burst size affected?

Please see our response to reviewer 1 above, who had the same question. Briefly, we conducted phage one-step growth experiments in liquid culture using both the AS strain and the slower-growing AR strains. Our results showed that there were no significant differences in the burst sizes among the strains. We now state this for the reader in the following lines of the revised manuscript.

Lines 185-188: “Furthermore, the burst sizes and latency periods of phage T6 also did not significantly differ among the AR strains, which provides further support that phage infection dynamics remained consistent despite the mutations (Extended Data Fig. 2).”

- along these lines, how do WT & mutant strains behave in a non-mixed colony? Maybe AR_{C,Tet} is generally more susceptible?

We thank the reviewer for this thoughtful comment. To address this concern, we conducted control experiments where we grew the AS strain and the AR_{C,Tet} strains separately (non-mixed) as monocultures on agar surfaces in the presence or absence of phage T6, as addressed in response to the earlier question about general susceptibility of phage. As described earlier, we observed a consistent decline in the diameter of the colony upon phage exposure

and didn't find that $AR_{C,Tet}$ is generally more susceptible to phage lysis. Figure 2c demonstrates that bacterial growth rate negatively correlates with the growth of phage-bacterial cocultures, where slower-growing mutants show diminished growth. We state this in the following lines of the revised manuscript.

Lines 183-185: “Despite their chromosomal mutations, both AR strains remained susceptible to phage T6, forming smaller biomass diameters on agar plates (Fig. 2c).”

- Fig. 4a (and S3), here it seems that even in the presence of phage, the whole fringe of the colony has lost the fluorescence. Is that an optical/microscopical artifact? Is this an appearance of escape mutants? In the simulations, the colonies do not show this appearance.

We thank the reviewer for this careful observation. Upon re-examination of the original images, we confirm that the apparent loss of fluorescence at the colony fringe in Figure 4a was indeed caused by an optical artifact, likely due to signal attenuation or image acquisition settings of the CLSM at the periphery of the biomass. We would like to emphasize that this was not the result of plasmid loss or the emergence of phage-resistant escape mutants. To resolve this issue, we have now repeated the imaging for the relevant samples and replaced the affected images with new representative images that do not exhibit the fluorescence loss artifact. The revised images are now included in the updated version of the manuscript.

Fig. 4: Phage lysis increases the persistence of slower-growing AR cells in the face of spontaneously generated AS cells. **a**, Representative CLSM images of strain $AR_{P,ChI}$ (magenta) in the absence (upper image) or presence (lower image) of phage T6. Non-fluorescent regions are composed of $AR_{P,ChI}$ cells that spontaneously lost plasmid pEF001 and became AS cells. We took the images after ten days of incubation at 21°C in the absence of antibiotic pressure. **b**, The proportions of the total biomass area occupied by strain $AR_{P,ChI}$ in the absence or presence of phage T6. **c**, The biomass diameters in the absence or presence of phage T6. **d**, The proportions of $AR_{P,ChI}$ cells as a function of the radial distance from the centroid of the biomass. For **b–d**, each data point is an independent experimental replicate ($n = 5$), the black data points are for experiments in the absence of phage T6, and the green data points are for experiments in the presence of phage T6. For **b,c**, the p -values are for two-sample two-sided Welch tests.

- when the authors score for loss of resistance bearing plasmids: Is this truly the loss of the plasmid or is it just a loss of fluorescence? How about the growth rate of 'dark cells' isolated from the colonies?

We thank the reviewer for this important question. To address this, we isolated the non-fluorescent 'dark cells' from the colony biomass and assessed both their antibiotic resistance phenotype and growth rate. Our experiments confirmed that these dark cells were unable to grow on chloramphenicol-containing agar plates, indicating that they had indeed lost the resistance-bearing plasmid pEF001 rather than simply experiencing fluorescence silencing. These results confirm that the loss of fluorescence we observed accurately reflects true plasmid loss and the associated release from the metabolic burden of plasmid maintenance. We now state this for the reader in the following lines of the revised manuscript.

Line 303-306: *"We confirmed that the non-fluorescent cells had indeed lost the resistance-bearing plasmid pEF001 rather than only the expression of the fluorescent protein encoding gene by verifying that they could no longer grow on agar plates amended with chloramphenicol."*

- If I understood correctly, the phages are added quite early to the nascent colony – does this really only affect cells growing at the fringes of this colony?

We thank the reviewer for this thoughtful question regarding the timing of phage addition and its spatial effects within the growing colony. Indeed, the phage was introduced after six hours of initial colony growth, at which point a visible biomass had formed but the colony was still in the early stages of expansion. While the phage is capable of infecting susceptible cells throughout the biomass, our experimental observations and prior theoretical work indicate that phage predation predominantly impacts cells located at the biomass periphery. This is due to mass transfer limitations within the dense interior of the colony, which restrict phage diffusion and access to interior cells. In contrast, the peripheral cells remain more exposed and are actively growing due to better nutrient access, making them more susceptible to phage infection. This spatially structured exposure is consistent with our imaging data showing preferential clearance of fast-growing cells at the colony edge, and it aligns with the established dynamics of phage-bacteria interactions in structured environments.

- Fig. 2a, lower panel, in the absence of phage, some AR_{C, Str} cells are retained, but seem to be completely absent in the presence of phages. How do the authors (and the simulations) explain this?

We thank the reviewer for this careful observation. Our model is unable to recapitulate these observations, but we suspect it is due to the relatively severe costs for streptomycin resistance by this strain (Fig. 2c). Under phage predation, these costs could be magnified, thus causing the complete displacement of resistant types from the system. However, due to the small effect size of this phenomenon (Fig. 2b) and because it does not affect our main conclusions, we chose not to further investigate this at this point. We nevertheless acknowledge that it could be an interesting avenue for future research.

Minor:

- maybe use a more contrasting color for the phages in Fig.1

We thank the reviewer for this helpful suggestion. We have revised Figure 1 by adjusting the size of the phage to enhance its visibility and more clearly distinguish it from the bacterial populations. The updated figure is now included in the revised manuscript.

Fig. 1: Schematic of the “peripheral kill-the-winner” hypothesis. In the absence of phage, we expect that faster-growing antibiotic sensitive (AS) cells (cyan) will displace slower-growing antibiotic resistant (AR) cells (magenta) along the biomass periphery. In the presence of phage, however, we expect that the slower-growing AR cells will persist with the faster-growing AS cells. This is because the faster-growing AS cells will disproportionately occupy the biomass periphery, and they will therefore be more susceptible to phage lysis. This will increase the removal of AS cells from the biomass and counteract the benefits of their faster growth relative to the slower-growing AR cells, thus increasing the persistence of the slower-growing AR cells.

- 107 ff, this part I would move to the results part, it’s not an introduction anymore.

We thank the reviewer for the suggestion. However, we believe that incorporating a brief overview to the experimental system in the introduction section is important, as it effectively bridges the knowledge gap identified in the introduction section and the results. We feel that this approach is not unconventional and provides essential context that prepares the reader for the detailed findings presented later. Thus, we respectfully maintain the current structure.

- 484, space after ,burden’.

Changed as recommended

Reviewer #3 (Remarks to the Author):

This paper presents a beautiful study of phage lysis on the edge of growing bacterial colonies and proves that this predation provides self-organized protection for the slower-growing bacteria. The study nicely combines experiment and modeling and is a delight to read.

We sincerely thank the reviewer for their positive and encouraging comments. We greatly appreciate the recognition of our combined experimental and modeling approach and are pleased that the study was found to be enjoyable to read.

My only comment concerns the formulation of the model. The authors define phage as a killing probability at the periphery of a colony. I guess the model was implemented as a killing rate, where bacteria on the periphery have a chance to be eliminated per unit timestep.

We thank the reviewer for this insightful comment and fully agree that our original wording may have caused confusion. In our individual-based computational model, the phage effect was indeed implemented as a per time step killing probability applied to peripheral cells. At each simulation time step, cells located at the colony periphery were assigned a probability of being lysed by phage, effectively modeling a killing rate dependent on the phage lysis probability parameter. To prevent such confusion, we have now changed the term “probability” to “rate” throughout the entirety of the manuscript.

An even smaller comment concerns the somewhat boring but very professional title of the manuscript. I liked the “Peripheral kill the winner” concept emphasized throughout the manuscript, and maybe the authors could consider integrating this into a more eye-catching title.

We appreciate the comment regarding the manuscript title and agree that highlighting the “Peripheral kill-the-winner” concept could improve the impact and visibility of the work. Based on this recommendation, we change the title to “Phage-mediated peripheral kill-the-winner facilitates the maintenance of costly antibiotic resistance in the absence of antibiotic pressure”

I recommend the publication of this manuscript.

We sincerely thank the reviewer for their positive assessment and recommendation for publication.

Reviewer #4 (Remarks to the Author):

This study investigates how lytic phages can modulate bacterial spatial organization to facilitate the persistence of antibiotic-resistant bacteria, even in the absence of antibiotic pressure. Using a combination of experimental and computational approaches, the authors propose and test the “peripheral kill-the-winner” hypothesis, which suggests that phages disproportionately lyse faster-growing antibiotic-sensitive bacteria at the biomass periphery, enabling slower-growing AR bacteria to persist. These findings have implications for understanding antibiotic resistance maintenance, eco-evo dynamics, and the application of phage therapy in bacterial population control. This is a well-executed and robust study, with strong experimental design and clear data presentation. However, my main concern is that the broader significance of this mechanism remains unclear. While the study convincingly demonstrates this effect under controlled experimental conditions, its relative importance compared to other mechanisms is not fully addressed. This uncertainty could limit the study’s impact, particularly in real-world microbial communities where multiple competing factors influence these dynamics. My comments below are thus mostly in this vein—focusing on clarifying how significant this mechanism is in the broader context of antibiotic resistance maintenance and microbial ecology.

We sincerely thank the reviewer for the positive assessment of our study. To address the reviewer's concerns, we have expanded the Discussion section of the manuscript to better contextualize our findings relative to other known mechanisms contributing to the persistence of antibiotic resistance in microbial communities. We also acknowledge the need for future studies to quantify the relevance of this mechanism in the face of other ecological and evolutionary drivers in more complex, multi-species, and environmentally variable systems. We have discussed these issues in the discussion as described below.

Comments

While the study discusses bacterial spatial organization, it does not explicitly address biofilms, which are crucial to understanding phage-bacteria interactions. Biofilms can provide protection against phage infection due to extracellular polymeric substances and structural heterogeneity. A discussion of how biofilm formation might alter the "peripheral kill-the-winner" mechanism would strengthen the ecological relevance of the study.

We thank the reviewer for this insightful suggestion. We fully agree that biofilms represent an ecologically and clinically relevant context for phage-bacteria interactions. While our study focuses on surface-associated bacterial colonies, which share certain spatial structuring features with biofilms, it does not explicitly address biofilm-specific factors such as extracellular polymeric substances (EPS) and increased structural heterogeneity. To clarify the ecological relevance of our findings, we have now expanded the Discussion section to consider how biofilm formation might modulate the peripheral kill-the-winner dynamics. Specifically, we discuss how EPS could limit phage diffusion and alter the accessibility of bacterial populations at the biofilm periphery, potentially modifying the selective pressures imposed by phage predation. We also consider how biofilm-associated phenotypic heterogeneity might influence the persistence of antibiotic-resistant subpopulations. This expanded discussion appears in the revised manuscript.

Lines 543-553: *"While our study focuses on surface-associated bacterial colonies, it is important to consider how these dynamics might extend to biofilm-forming communities, which are prevalent in natural and clinical settings. Biofilms exhibit distinct structural and physiological properties that could modulate phage-host interactions. The extracellular polymeric substances (EPS) matrix in biofilms can impede phage diffusion, potentially restricting access to bacterial populations at the periphery and altering the strength of peripheral kill-the-winner selection. Additionally, biofilm-associated heterogeneity, including the presence of metabolically inactive persister cells, could further influence the persistence of antibiotic-resistant subpopulations^{61,62,63}. Future studies incorporating well-characterized biofilm models and assessing the impact of EPS composition and biofilm maturation on phage predation dynamics are important to fully understand the ecological significance of this mechanism in biofilm systems."*

The paper convincingly demonstrates the effect of phage predation on AR persistence in specific experimental conditions. However, it is difficult to anticipate the relative importance of this mechanism compared to other well-documented resistance maintenance strategies, such as compensatory mutations, plasmid persistence, and co-selection with other traits. A discussion on whether this mechanism is likely to play a dominant or minor role in natural environments would be valuable. Similarly, the study uses only one bacterial strain and one phage, under highly controlled conditions. While this simplicity allows for clear conclusions, it raises concerns about generalizability.

We sincerely thank the reviewer for this thoughtful comment. We fully agree that understanding the relative importance of the peripheral kill-the-winner mechanism compared to other resistance maintenance strategies is crucial for assessing its ecological relevance. In response, we have expanded the Discussion section to explicitly address this point. We now compare our mechanism with established processes such as compensatory evolution, plasmid stabilization mechanisms, and co-selection with other stressors, highlighting that the relative contribution of peripheral kill-the-winner dynamics will likely be context-dependent. We also acknowledge the limitations of our study in terms of generalizability. The use of a single model bacterial strain and phage system under controlled laboratory conditions was intentional to allow for mechanistic dissection. However, we agree that future work should explore diverse phage-host pairs, broader environmental conditions, and multispecies communities to fully evaluate the prevalence and strength of this mechanism in natural systems. These points have now been added to the revised Discussion section.

Lines 555-567: *“The reductionist approach we employed in this study provides a powerful means to identify and isolate mechanisms but is limited in its generalizability. Our study is based on a simplified system involving a single bacterial species and a single phage, where the phage was introduced at a single point in time. While this approach enabled us to dissect the mechanism in detail by minimizing confounding factors, it does not capture the complexity of natural environments. In nature, phage-host dynamics can vary widely depending on host genetic background, phage infection strategies, and the presence of microbial competitors. Moreover, the timing of phage addition likely plays an important role in determining how phages modulate the spatial structure of biofilms, especially as diffusion limitations become more pronounced with increasing biomass. Future research should explore the extent of peripheral kill-the-winner effects across diverse bacterial and phage systems, in more complex environments, and within multispecies microbial communities to fully assess the ecological and evolutionary significance of this mechanism.”*

The authors created resistant strains through experimental evolution. However, it is unclear whether multiple independent resistant strains were tested or if conclusions were drawn from a single resistant colony after serial passage. Clarifying what the biological replicates represent would help in assessing the robustness of the findings. If a different resistant colony would've been picked, would the conclusions be different?

We thank the reviewer for this important question regarding the experimental design and the definition of biological replicates. We confirm that for both chromosomally resistant strains, we selected a single representative resistant colony following experimental evolution and verification of resistance. This clone was then used to establish frozen stocks, which served as the standardized starting point for all subsequent experiments. In our surface-associated competition assays, biological replicates represent independent colony outgrowths initiated from these frozen stocks. We fully acknowledge that different resistant clones might harbor distinct mutations with potentially variable fitness costs. However, our study is designed to test the basic principle that phage predation modulates competition between strains differing in growth rate due to resistance costs. Importantly, our individual-based computational model, which is independent of any specific genetic change or resistance mechanism, recapitulates the key dynamics observed experimentally, where the model demonstrates that the peripheral kill-the-winner mechanism is robust across a range of fitness costs. Therefore, while the exact quantitative outcomes could vary with other resistant clones, we are confident that our main conclusions remain qualitatively valid regardless of the genetic or mechanistic basis of resistance.

Reviewer #5 (Remarks to the Author):

We sincerely thank the reviewer and the Early Career Researcher for their thoughtful evaluation and constructive feedback. We greatly appreciate the contributions to improving the quality of our manuscript.